# Multiplexed computations in retinal ganglion cells of a single type

Stéphane Deny[1,3], Ulisse Ferrari[1], Emilie Macé[1,4], Pierre Yger[1], Romain Caplette[1], Serge Picaud[1], Gašper Tkačik [2] & Olivier Marre [1]

In the early visual system, cells of the same type perform the same computation in different places of the visual field. How these cells code together a complex visual scene is unclear. A common assumption is that cells of a single-type extract a single-stimulus feature to form a feature map, but this has rarely been observed directly. Using large-scale recordings in the rat retina, we show that a homogeneous population of fast OFF ganglion cells simultaneously encodes two radically different features of a visual scene. Cells close to a moving object code quasilinearly for its position, while distant cells remain largely invariant to the object's position and, instead, respond nonlinearly to changes in the object's speed. We develop a quantitative model that accounts for this effect and identify a disinhibitory circuit that mediates it. Ganglion cells of a single type thus do not code for one, but two features simultaneously. This richer, flexible neural map might also be present in other sensory systems.

[1] Institut de la Vision, INSERM UMRS 968, UPMC UM 80, CNRS UMR 7210 Paris, France. [2] Institute of Science and Technology Austria, 3400 Klosterneuburg, Austria. [3] Present address: Neural Dynamics and Computation Lab, Stanford University, CA, 94305, USA. [4] Present address: Neural Circuit Laboratories, Friedrich Miescher Institute for Biomedical Research, Maulbeerstrasse 66, 4058 Basel, Switzerland. Correspondence and requests for materials should be addressed to O.M. (email: olivier.marre@inserm.fr)

A major challenge of the visual system is to extract meaningful representations from complex visual scenes. Feature maps, where the same computation is applied repeatedly across different sub-regions of the entire visual scene, are essential building blocks for this task, for both sensory networks[1,2] and artificial vision systems[3]. Ganglion cells, which form the retinal output, can be divided into different types[4–7]. In the classical view of retinal function, cells of the same type extract a single feature from the visual scene and generate a feature map that is then sent to the brain[8]. This "one type = one feature" view is well illustrated in the retina when objects move across the visual field at constant speed. In this case, previous work has shown that a single-type indeed represents a single feature of the scene[9–12].

However, processing by ganglion cells also depends on the visual context[13–17], so that feature extraction will be influenced by the global parameters of the visual scene, e.g., by its luminance and contrast. Furthermore, ganglion cell activity can be modulated by stimulation outside of the cells' classically defined receptive fields[18–22], implying that feature extraction may not be entirely local, especially when presented with complex, dynamical stimuli. As a result, it is not clear how irregular trajectories of moving objects, which are ubiquitous in natural scenes[23,24], are represented by ganglion cells of the same type.

Here we show that a single-ganglion cell type extracts simultaneously two very different features from a visual scene composed of irregularly moving bars. Within a homogeneous population of fast OFF ganglion cells recorded simultaneously, cells whose receptive field center overlaps with an object perform a quasilinear computation that is highly sensitive to the position of the object. In contrast, cells of the same cell type that are far from any moving object respond more nonlinearly to fast motion, and are largely invariant to the exact position of distant objects. Individual cells switch from this computation to the other when their receptive field center is stimulated. We constructed a model that quantitatively accounted for these findings, and determined that the observed scheme of distal activation is implemented by a disinhibitory circuit of amacrine cells.

## Results

**Ganglion cells respond to distant moving objects**. We recorded large ensembles of ganglion cells from the rat retina using a micro-electrode array of 252 electrodes[25,26]. We measured the receptive field center of each cell with binary checkerboard noise. To separate ganglion cells into different types, we displayed several stimuli (full field flicker, drifting textures) and grouped together cells with similar responses (Methods section). In the following, we focus on a single group composed of well-isolated fast OFF cells. Their responses to spatially uniform stimuli were nearly identical (Fig. 1a), and their receptive fields clearly tiled the visual space (Fig. 1b). Their response to a full field flash was transient, and there was only a response to a light decrease, not to a light increase (Supplementary Fig. 1). For this reason the type studied here corresponds most likely to OFF alpha transient cells (8a or 8b in Baden et al.[7]). When comparing to previous classifications performed on ganglion cells based on anatomy, this type most likely corresponds to the G3, G7, G11, or G18 types described by Volgyi et al.[27].

We then displayed a bar of width 100 μm moving randomly over the visual field. This dark bar over a gray background was animated by a Brownian motion with a feedback force to keep the bar positioned over the array. As expected, ganglion cells whose receptive field center overlapped with the bar position responded reliably to a repeated trajectory, as shown by their peristimulus time histogram (PSTHs) in Fig. 1c. More surprisingly, reliable responses were also elicited in cells whose receptive field centers

were far away from the bar. The receptive field center diameter was on average 287 ± 23 μm (mean ± SD, n = 25), and cells as far as 670 μm from the closest bar position responded to the moving bar.

These distant cells fired synchronously in response to the moving bar, largely independently of the location of their receptive field, while central cells did not. Central cells were only synchronous when they were very close to each other. The mean cross-correlation between the responses of pairs of central cells was 0.02 ± 0.04 (mean ± SEM, n = 20 pairs, Pearson correlation r) for cells separated by more than 200 μm along the axis perpendicular to the bar. In comparison, distant cells remained synchronous over large distances (Fig. 1d). The mean cross-correlation was 0.53 ± 0.03 (mean ± SEM, n = 35 pairs) for distant cells separated by more than 200 μm.

This distant activation had a profound effect on the structure of the retinal activity: while the bar moved within a region around 0.4 mm wide, ganglion cells were activated over an area wider than 1.4 mm (Fig. 1e).

**Two computations within a single type of ganglion cells**. We asked if the observed ganglion cell responses to motion outside their receptive field centers could be explained by standard models of the retina. We fitted a Linear-Nonlinear-Poisson (LN) model[28–31] (Fig. 2a) to the response of each cell to non-repeated trajectories of the moving bar. To test it, we repeated the same bar trajectory 54 times and compared predictions of the model with the measured PSTH for each cell. When the bar was moving close to or inside the receptive field center of the cell, the LN model predicted very well the response to the repeated sequence (r = 0.79 ± 0.02, n = 25, Fig. 2b). However, for cells that were distant from the bar, the same LN model failed at predicting their responses (r = 0.12 ± 0.02, n = 19, Fig. 2c). Performance was much lower (p ≤ 10^−5, two-sample t-test, t = 23), and could not be explained by a decrease in the reliability of the response (the ratio of explainable variability predicted by the model was 13 ± 2%, n = 19, see Methods section). This low performance was obtained despite the fact that we fitted the LN model directly on the responses to the distant bar. The low performance was also not due to any intrinsic property of these cells, but was related to the distance between the bar and the receptive field. When we displayed the moving bar in different locations, the same cells that were previously not predicted by the LN model (r = 0.12 ± 0.02, n = 19), with RFs far from the bar, were predicted very well by a LN model when the bar was displayed inside their receptive field center (r = 0.79 ± 0.02, n = 19; p ≤ 10^−5, paired-sample t-test, t = 19.9). In summary, the LN model was a good model for a bar moving inside the receptive field center, but not outside. We observed these distant responses that could not be explained by a LN model over a broad range of luminances (Supplementary Fig. 2).

To improve the prediction, we considered a subunit model, with two stages of processing that implements a nonlinear summation[30–34] (Fig. 2d).

In the first stage, ON and OFF subunits tiled the visual space, and convolved the stimulus with a linear filter followed by a rectification. Their output was pooled linearly and rectified to predict the firing rate (see Methods section for details).

This model predicted very well the responses of distant ganglion cells to a repeated random trajectory (r = 0.73 ± 0.02, n = 19 Fig. 2d, e). Performance was high for all distances of the receptive field to the bar (Fig. 2f), demonstrating that the subunit model robustly captured responses that were not predicted by the LN model.

Note that this subunit model performed also well for center stimulation: in this case, rectified subunits were summed in the

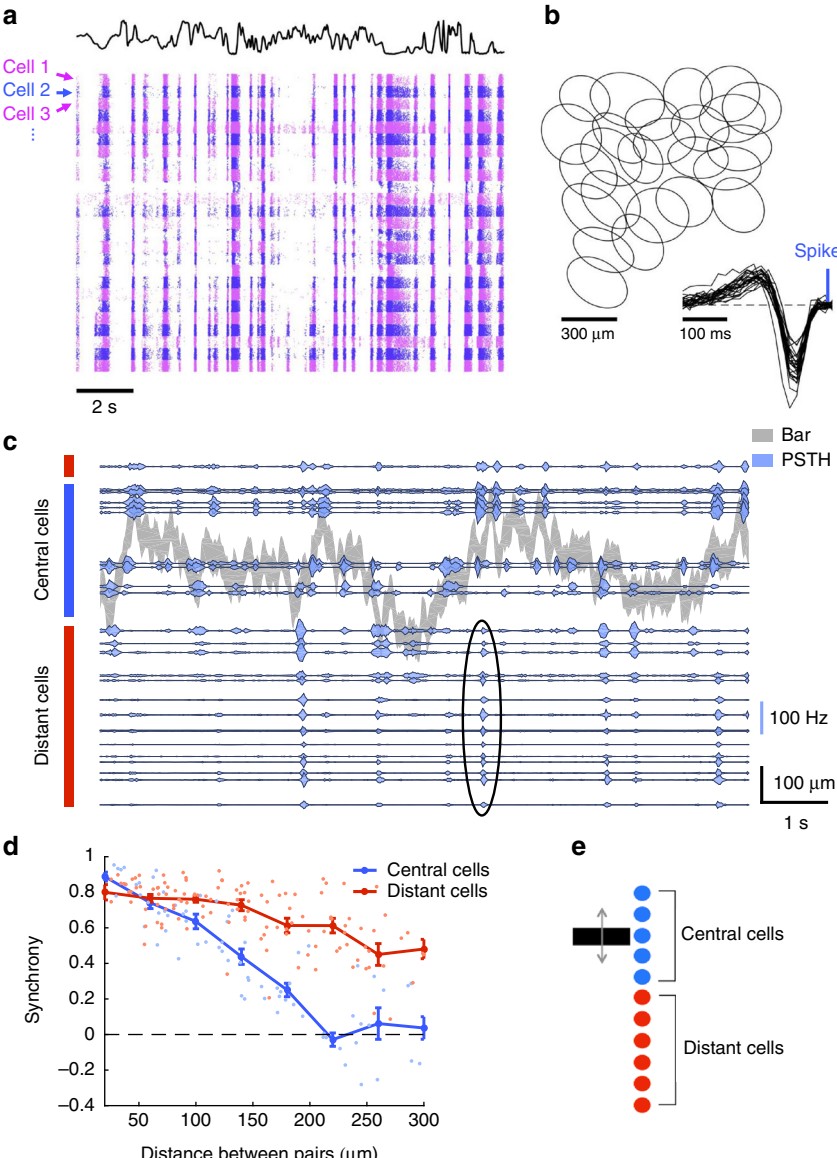

**Fig. 1** A single-cell type responds synchronously to distant moving objects. **a** Raster of 25 cells of the same type responding to a full field uniform flicker. Each line corresponds to a repeat of the stimulus, and each cell is indicated by a different color (alternating pink and blue). The black curve indicates the light intensity of the flicker over time. **b** Receptive fields of a population of ganglion cells of the same type. Each ellipse represents the position and shape of the spatial receptive field associated with one cell (1-SD contour of the 2D Gaussian fit to the spatial profile of the RF). Inset: temporal profiles of the receptive fields of the same cells. **c** PSTHs of multiple ganglion cells responding to repeated presentations of a randomly moving bar. Gray shade: position of the bar as a function of time (shade width corresponds to the bar width). Blue traces: PSTHs of individual ganglion cells, with baselines positioned to scale relative to the bar. Blue and red vertical rectangles indicate central and distant cells, respectively. Black ellipse shows an example synchronous firing event of the distant cells. **d** Cross-correlation between PSTHs of pairs of cells, as a function of their pairwise distance measured along the bar motion axis, shown separately for cells whose receptive field center either was (blue) or was not (red) stimulated by the bar. Curves: average values ± SEM. **e** Schematic diagram shows central cells (blue) and distant cells (red) that respond synchronously

second stage of the model with opposite signs for ON and OFF subunits (Supplementary Fig. 8a, b, respectively). This "push–pull" organization linearized the net effect of the stimulus[35].

To test the generality of our model, we displayed a randomly moving texture. For this more complex stimulus, the cell responses were also accountable by a LN model in the center and by a subunit model for distant stimuli (Supplementary Note 1, Supplementary Fig. 3).

Our results showed that a population of cells of the same type extracted simultaneously two features from a single-moving object. Cells whose receptive field centers overlapped with the object performed a quasilinear computation on the stimulus, i.e., a computation well recapitulated by a LN model. Distant cells performed a nonlinear computation that was captured by a more complex model described above, the subunit model. Therefore responses to a distant moving bar could not be simply explained by using a broader linear filter within the LN model framework. Taken together, these findings show that two radically different computations, performed on the same stimulus, can co-exist within a population of ganglion cells of a single type (Fig. 2g).

Note that the coexistence of these two different computations is not an ubiquitous feature of ganglion cells. We analyzed another type of retinal ganglion cells in our data set whose receptive fields

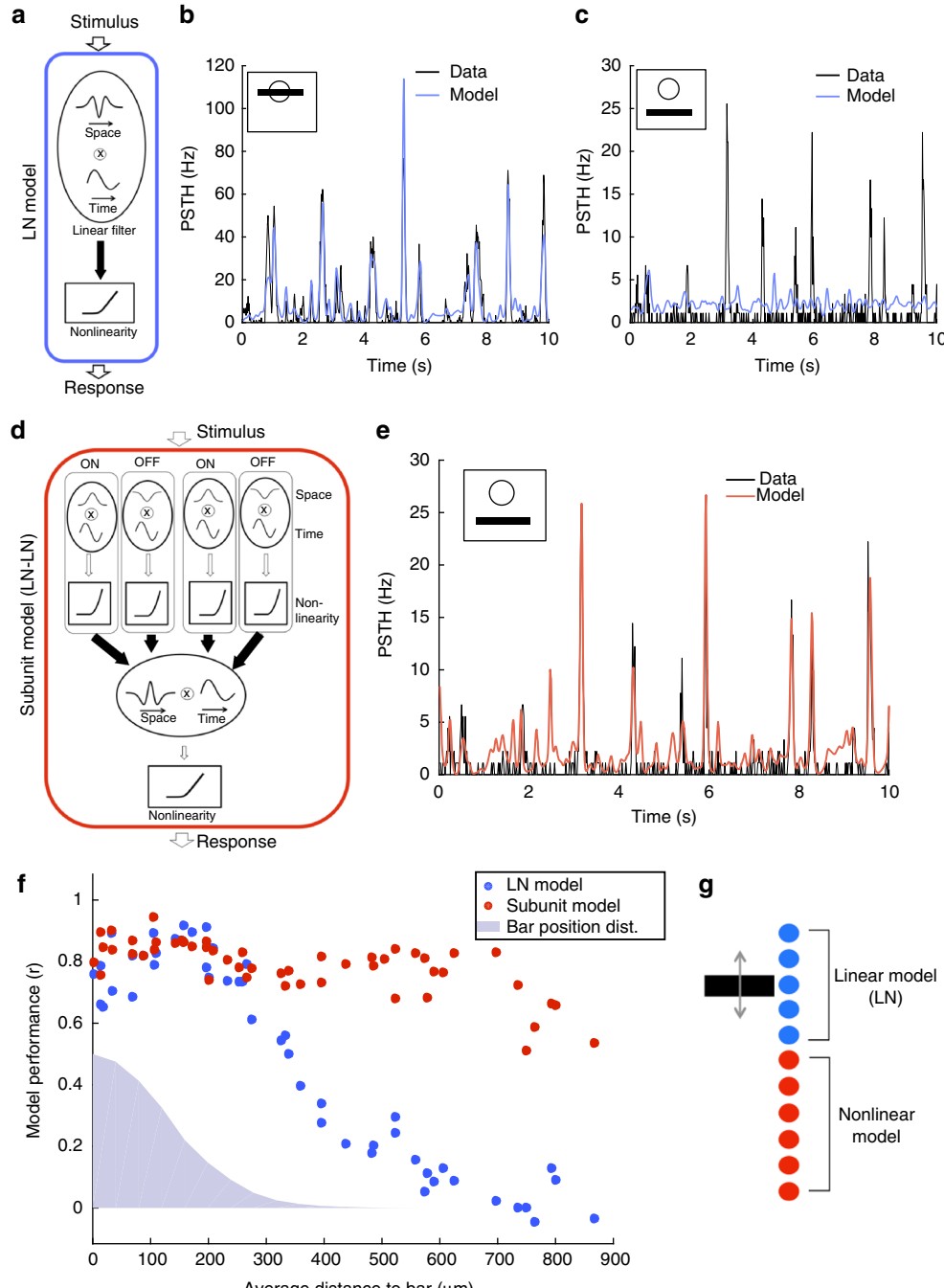

Fig. 2 OFF ganglion cells perform a quasilinear computation in their receptive field center, and a nonlinear computation in the surround. **a** Schematic of the LN model, composed of a linear filter and static nonlinearity. **b** Response (PSTH, black) of a ganglion cell whose receptive field center is stimulated by the bar, is predicted by the LN model (blue). r = 0.89. **c** Response (PSTH, black) of the same ganglion cell when the bar is far from the receptive field center, is not predicted well by the LN model (blue). r = 0.02. **d** Schematic of the subunit model, composed of a first stage (each subunit linearly filters the stimulus and applies a static nonlinearity), followed by weighted linear pooling and a second nonlinearity. Filled arrows correspond to learned weights of the model. **e** Response (PSTH, black) of the same ganglion cell (as in **b**, **c**) to distant stimulation is predicted well by the subunit model (red). r = 0.83. **f** Performance of the LN (blue) and subunit (red) models in predicting ganglion cell responses, as a function of the distance of the cell to the bar. Blue shade: position distribution of the bar. **g** Schematic showing that cells whose receptive field center is on top of the moving bar perform a quasilinear computation while distant cells perform a nonlinear computation

clearly tiled the visual field (Fig. 3g): namely, ON cells that probably correspond to ON transient type 18a or 18b in Baden et al.[7], as they responded transiently to an ON full field flash but not to an OFF one. In their case, the LN model failed to predict the response even for a stimulation of the center (Fig. 3a, b), and

also for a distant stimulation (Fig. 3a, c). The subunit model predicted quite well the responses to both central and distant stimulations (Fig. 3d–f, h). For this type, the computations performed for central and distant stimulations are both nonlinear. This is in contrast with the OFF type, where two

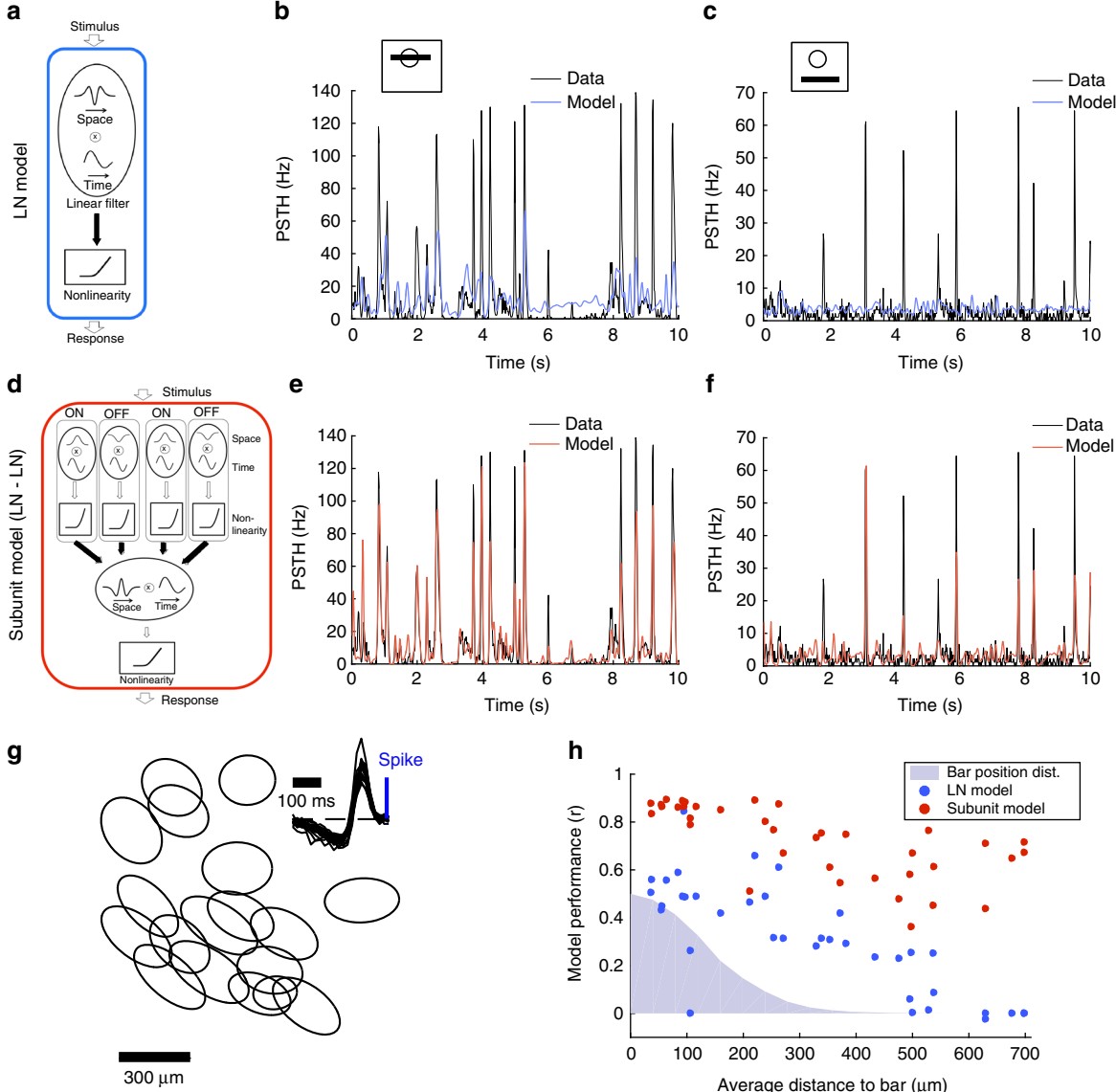

**Fig. 3** Responses of ON ganglion cells to the randomly moving bar. **a** Schematic of the LN model, composed of a linear filter and static nonlinearity. Filled arrows correspond to learned weights of the model. **b** Response (PSTH, black) of a ganglion cell whose receptive field center is stimulated by the bar, is not well predicted by the LN model (blue). r = 0.56. **c** Response (PSTH, black) of the same ganglion cell when the bar is far from the receptive field center, is not predicted well by the LN model (blue). r = 0.01. **d** Schematic of the subunit model, composed of a first stage (each subunit linearly filters the stimulus and applies a static nonlinearity), followed by weighted linear pooling and a second nonlinearity (see Methods section for details). Filled arrows correspond to learned weights of the model. **e** Response (PSTH, black) of the same ganglion cell (as in **b**, **c**) to central stimulation is predicted well by the subunit model (red). r = 0.9. **f** Response (PSTH, black) of the same ganglion cell (as in **b**, **c**) to distant stimulation is predicted well by the subunit model (red). r = 0.77. **g** Receptive fields of a population of ON ganglion cells of the same type. Each ellipse represents the position and shape of the spatial receptive field associated with one cell (1-SD contour of the 2D Gaussian fit to the spatial profile of the RF). Inset: temporal profiles of the receptive fields of the same cells. **h** Performance of the LN (blue) and subunit (red) models in predicting ganglion cell responses, as a function of the distance of the cell to the bar. Blue shade: position distribution of the bar

computations of different nature are performed on central and distant stimulations, a quasilinear one in the center and a nonlinear one for distant stimuli.

**Switching between two modes of computation.** Since the cells perform distinct computations in their center and in their distant surround, we studied how these computations interact when both center and surround are stimulated at the same time. What happens to distant cells if another bar is simultaneously shown inside their receptive field center? One possibility is that central and distant responses are simply added, so that the response to

two moving bars would be the sum of the responses to each bar presented separately.

To test this, we displayed two bars moving randomly, with distinct trajectories, in two different locations. The distance between the bars' average positions was 600 μm. We also displayed each bar in isolation, at the same respective locations and animated by the same trajectory as in the combined bar stimulus. We found that the response to the two bars was not equal to the sum of the individual responses to each bar presented separately (Fig. 4a). Instead, if a bar was moving inside the receptive field center of the cell, the response to the distant bar was suppressed, while the distant bar exerted a negligible effect on

the response to the central bar. Specifically, when one of the bars was moving inside the receptive field of a cell, the response to the combined bar stimulus was highly similar to the response to the single-central bar ($r = 0.91 \pm 0.01$, $n = 13$). The residual response to the distant bar in the presence of simultaneously presented central motion correlated poorly with the response to the distant bar alone, and this discrepancy could not be explained by noise (Fig. 4b, see Methods section for details—note that a similar suppression was observed for the ON type described above (Supplementary Fig. 4)).

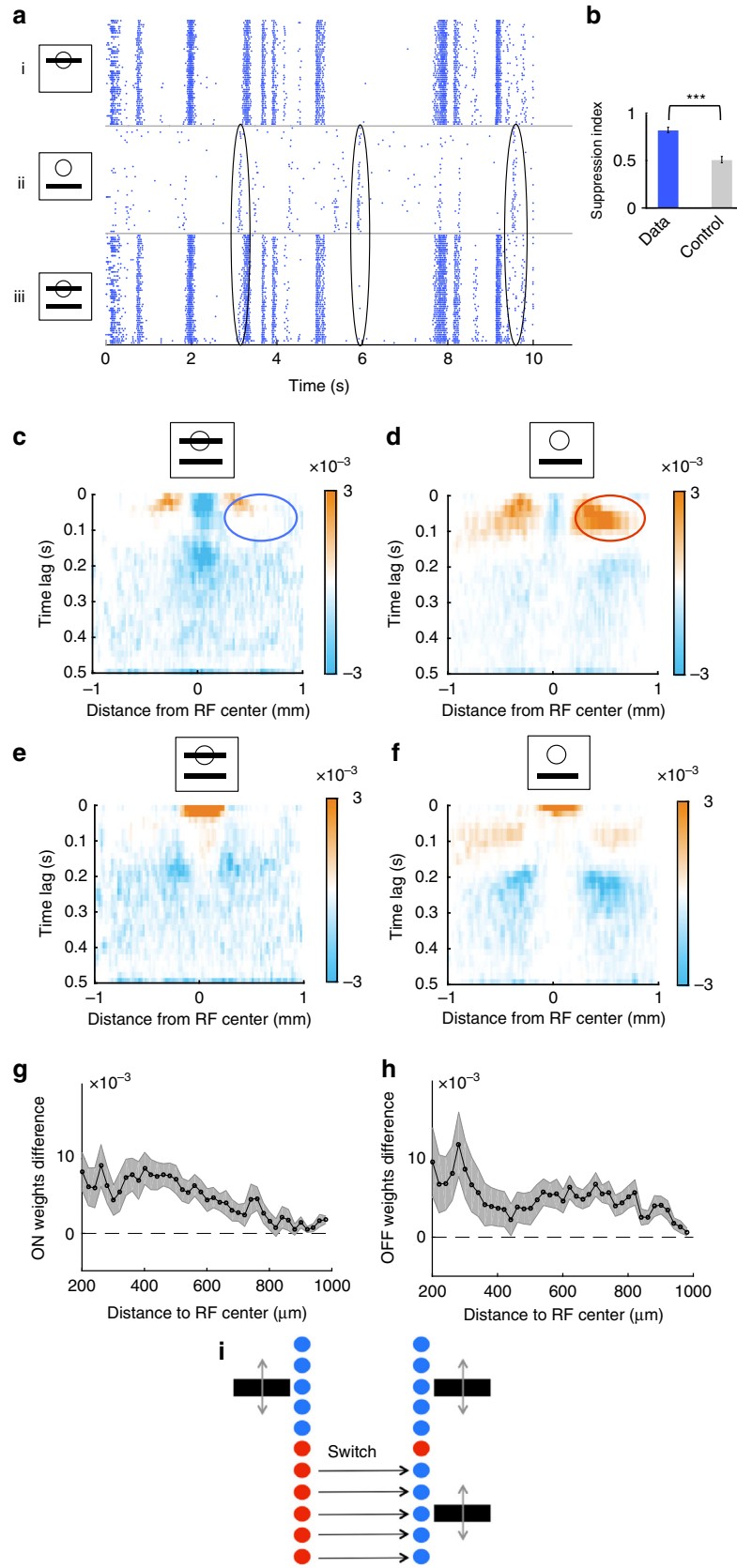

To quantify further the observed suppression, we fitted the subunit model for the three stimuli separately (bar 1, bar 2, two bars). We averaged the inferred subunit weights for all distant cells to obtain an "average cell" and understand better how this cell type pools stimulation from the far surround (Fig. 4c–f; see Methods section). We focused on distant subunits here because, in the single-distant bar condition, the bar spends very little time inside the receptive field center of the cell, making it difficult to estimate the subunit weights inside the receptive field in this condition. For both ON and OFF subunits, between 0 and 200 ms lag, the distant subunit weights decreased in the combined bar condition compared to the single distant bar condition. To quantify this, we subtracted the subunit weights between the two conditions (Supplementary Fig. 5) and averaged them between 0 and 200 ms. The subunit weight difference was always positive (Fig. 4g, h). Consequently, stimulation in the receptive field center decreased the contributions of distant subunits.

In summary, we observed a switch between two very different computations: cells changed from performing a nonlinear computation on distant stimuli to performing a quasilinear computation on the stimuli inside their receptive field centers (Fig. 4i).

The quasilinearity of the computation in the center is explained by the fact that the subunit weights of OFF and ON subunits in the center are organized in a "push–pull" manner, where positive weights for OFF subunits correspond to negative weights for ON subunits. In contrast, we did not find this push–pull organization of subunit weights for distant subunits, which probably explains why the distant computation could not be accounted by a LN model.

**Gain control explains the suppression of distant responses**. Our results indicate that the influence of distant inputs is suppressed when the receptive field center is stimulated. We have demonstrated that the suppression cannot be accounted for by a simple linear summation of the output of the subunits in the surround with the output of the linear filter in the center followed by a nonlinearity (Supplementary Note 2, Supplementary Fig. 6).

To elucidate further how central inputs suppress distant ones, we asked if the suppression increases gradually as central inputs become progressively stronger, or if the suppression is only activated once the strength of central inputs exceeds a threshold. To test this we displayed a series of stimuli where two bars were oscillating over the visual field at incommensurable frequencies (see Methods section). By averaging over the oscillation period of each bar, we could isolate the responses due to each bar. Our analysis focused on neurons for which one of the bars was within the receptive field center, while the other bar was outside. The central bar was displayed at several luminances, ranging from zero contrast (i.e., at background gray level) to maximally dark bar. We observed that responses to the distant bar decreased gradually as the luminance of the central bar went from gray to full dark, implying that the suppression of distant inputs was gradual (Fig. 5a, b). Next, we looked for a general model that

could explain center-strength-dependent suppression of responses to distal stimulation.

We hypothesized that the observed suppression is due to a gain control acting on the ganglion cell. In this view, distant inputs originating in the far surround are much weaker than the inputs originating in the center, and a gain control mechanism normalizes the cell's firing rate by the total overall input. Specifically, our model sums the inputs coming from central and distant stimulation, averages the result over a long (1 s) temporal window to get the normalization signal, and finally divides the instantaneous input by this normalization to get the final firing rate prediction (see Methods section). When the center is stimulated, the gain control will thus divide the output by a large normalization factor, which will suppress weak inputs from the surround (Fig. 5c). However, when the surround is stimulated alone, the gain control will act as an amplifier, allowing the cell to respond to the distant bar (see Fig. 5d, e for an illustration).

We fitted such a gain control model, inspired by Shapley and Victor[36], Berry et al.[9] to neurons stimulated by two bars with different luminances (see Methods section). The model predicted very well the responses to the different stimuli (Fig. 5a): it explained $84 \pm 3\%$ ($n = 168$, see Methods section) of the variance in responses to the distant bar across all contrast conditions.

These results demonstrate that a gain control is at work in these ganglion cells, and that this gain control can explain the observed suppression of distant responses. In summary, the subunit model equipped with gain control represents a functional model to explain all the results described above. The gain control can be implemented in several ways, which are not mutually exclusive[37]. One possibility is that gain control is shaped by the intrinsic properties of ganglion cells, and in particular the slow inactivation of sodium channels[38]. Another possibility is a feed-forward inhibition from amacrine cells[39].

**Position coding and change detection within a single type**. Our model showed that fast OFF ganglion cells performed two very different computations on the stimulus: a quasilinear one (LN) inside their receptive field center and a nonlinear one outside. But what visual feature is extracted by each of these computations? To address this question, we first plotted the distribution of bar positions 100 ms before a spike. For an example central cell (Fig. 6a), this distribution was narrow and had a cell-dependent preferred location, indicating the ability of central cells to code for the position of the bar. In contrast, for a distant cell the same distribution remained broad and largely overlapping with prior distribution of bar positions (Fig. 6b), suggesting that distant cells were largely insensitive to the exact position of the stimulus. This lack of sensitivity to the position of the bar was confirmed at the population level: when we performed linear decoding on distant ganglion cells[22] to predict the position of the bar, the performance was much lower than when decoding central cells (Fig. 6e). Since the decoding performance increased with the number of cells, it is likely that, if we were to decode from an

**Fig. 4** Suppression of distant responses by a bar moving inside the receptive field center. **a** Response raster of a single cell to a moving bar presented in and/or outside of its receptive field center. Each dot is a single spike from the recorded cell. Each line corresponds to a different repetition of the same stimulus. (i) Bar moving inside the receptive field center. (ii) Bar moving outside the receptive field center. (iii) Both bars displayed together. Black ellipses indicate examples where the response to the distant bar is strongly suppressed by the stimulus inside the receptive field center. **b** Suppression index for real cells (data, blue) and due to noise (control, gray); see text and Methods section for details. Data are represented as mean ± SEM ($n = 13$ cells). The three stars indicate that the $p$-value of a two-sample $t$-test was lower than 0.001. **c, d** Spatio-temporal distribution of the ON-subunit weights for the second stage of the subunit model, averaged over all cells. **c** Combined bar stimulus. **d** A single-distant bar stimulus. Blue and red ellipses show the reduction of the weights in the surround in the presence of central bar motion. Since we focus on distant weights here (see text), central weights are saturated in these panels. **e, f** Same as **c, d** for OFF-subunits. **g** ON subunit weight difference between the single-distant bar stimulus and the combined bar stimulus, averaged between 0 and 200 ms time lag. The shades represent SEM. **h** Same as **g** for OFF-subunit weights. **i** Schematic showing how cells switch the mode of computation when a bar is displayed within their receptive field center

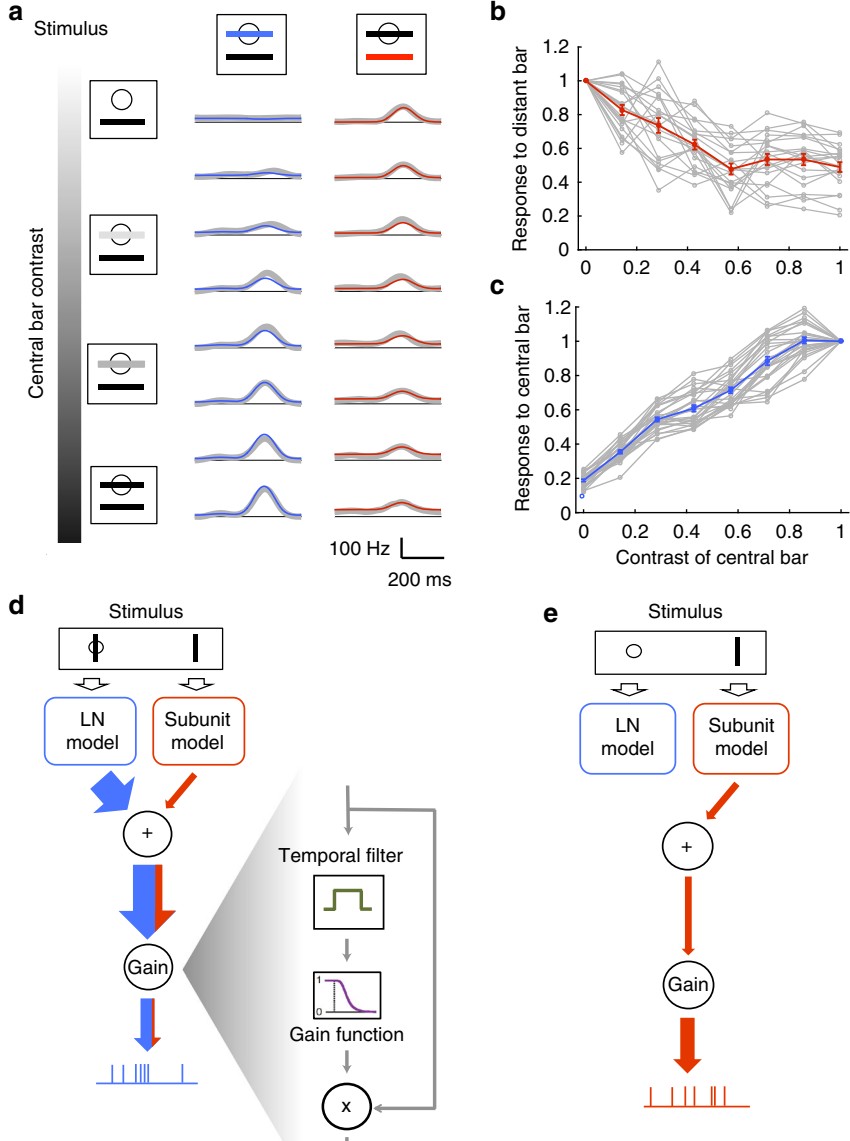

**Fig. 5** A global gain control model explains gradual suppression of distant responses. **a** Gradual suppression of the response to the distant bar as central bar contrast increases. Left: schematic of the stimulus. The central bar contrast increased progressively from top to bottom. Middle: response of one cell to the central bar for eight different central bar contrasts; real PSTH (gray) and prediction of the gain control model (blue). Right: response of the same cell to the distant bar as contrast of the central bar increases; real PSTH (gray) and prediction of the gain control model (red). **b** Peak of the response to the distant bar (normalized by the response to the distant bar alone) as a function of the contrast of the central bar for individual cells (gray) and on average (red). Error bars represent SEM. **c** Peak of the response to the central bar (normalized by the response to the central bar alone) as a function of the contrast of the central bar for individual cells (gray) and on average (blue). Error bars represent SEM. **d**, **e** Schematic of the effect of gain control. In case of a central stimulation **d**, a gain control component will suppress the output; weak inputs from the surround will barely modulate the response. In case of no central stimulation **e**, the gain control component will enhance the weak inputs corresponding to the distant bar

extremely large population of distant cells, we could recover the position of the bar, but this would require many more cells than when decoding from central cells.

Distant cells nevertheless were selective for the stimulus. The average speed (absolute velocity) preceding the spike of a distant cell showed a preference for fast bar motions (Fig. 6d). By quantifying the information carried by each cell about bar position and speed (see Methods section, $n = 25$ cells), we confirmed that distant cell responses encoded substantially more information about speed than about position, whereas central cells coded primarily for position (Fig. 6f) but were also sensitive to speed (Fig. 6c). The observation of highly synchronous responses of distant cells to a random repeated bar trajectory

(Fig. 1) further supports our interpretation that the distant computation is largely invariant to the exact bar position. We also analyzed the subunit model fitted to cell responses in the previous section and found that cells close to the bar were much more sensitive to the bar position than distant cells, confirming these results (Supplementary Note 3, Supplementary Fig. 7).

Cells close to the bar were thus sensitive to the exact position of the bar, while distant cells were largely invariant to the bar position (Fig. 6g). How can we interpret this invariance to position and simultaneous sensitivity for high-speed motion? Fig. 4d shows that cells pooled the output of distant subunits over a large spatial region of the surround. This pooling was largely unselective for position and thus explained how the observed

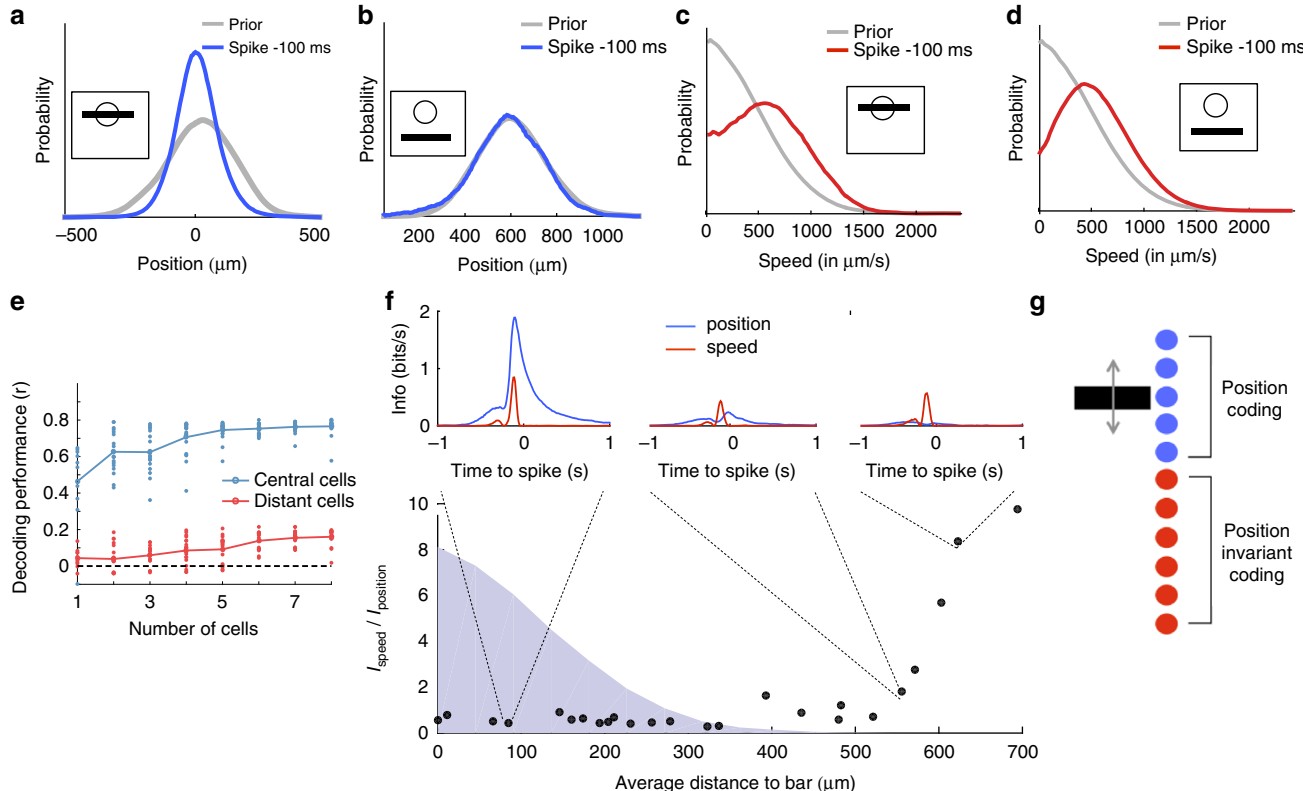

**Fig. 6** Central computation codes for position, while distant computation is invariant to position and codes for stimulus change. **a** Distribution of the bar positions for the complete stimulus trajectory ("prior distribution", gray) and 100 ms before the spike of a central cell (blue). Zero corresponds to the location of the cell's receptive field (RF) center. **b** Same as A for a distant cell with its receptive field center far from the bar. **c** Distribution of the absolute speed of the bar for the complete stimulus trajectory (gray) and 100 ms before the spike of a central cell (red). **d** Same as **c** for a distant cell (red). **e** Decoding performance of the position of the bar using linear decoding, against number of cells, for either central cells (blue) or distant cells (red). Each point corresponds to one subset of cells, and the curves correspond to the average performance. **f** Ratio between the information individual cells carry about bar speed vs about bar position, as a function of the average distance to the bar. Distribution of bar positions is shown as a blue shade. For selected cells, the insets indicate the mutual information between the spiking response and the position (blue) or the speed (red) at different time delays. **g** Schematic showing that central cells code for bar position while distant cells are nearly invariant to it

distant responses could remain nearly invariant to the position of the bar. To trigger a response in distant cells, the bar had to sweep across a large region of space: this would lead to an activation of a large number of subunits in a short amount of time that are summed together to result in the activation of a ganglion cell. This activation should only occur when the bar moves at sufficiently high speed, explaining the preference for high speed motion.

However, our model further predicted that flashing a large object in the cell's surround should also activate many subunits at the same time and trigger a response. We confirmed this was indeed the case (Fig. 7e; note that distant responses are ON responses, and this is also what the subunit model predicted). This result suggests that distant cells should not be viewed narrowly as encoding "high speed" (an interpretation that is natural for the moving bar stimulus). Besides, central cells were also sensitive to speed. Speed is therefore not exclusively coded by distant cells. We have not found any specific feature that would be only coded by distant cells, and not by central ones. Rather, a generic interpretation is that distant cells code for any large change in the stimulus, and are more strikingly characterized by their invariance to position than by their selectivity to a particular component of the visual scene. Note however that this is not a "perfect" change detector as it is not symmetric in polarity: these cells responded to large ON flashes in the far surround, but not to OFF flashes.

In summary, central cells are position sensitive, while distant cells are largely insensitive to the exact position of the stimulus and behave more like "change detectors".

**A disinhibitory circuit of amacrine cells for distant inputs**. We next examined how the computations required by our phenomenological model could be implemented by the retinal network. The subunits of our model may correspond to bipolar cells[40–42], although the exact correspondence between subunits and bipolar cells cannot be proven. For subunits in close physical proximity to the ganglion cell, the weights can result from direct synaptic connections between bipolar cells and the ganglion cell. In addition to these proximal connections, however, our model suggested that the ganglion cell also integrated the outputs of distant subunits, albeit with a smaller weight. What could be the circuit basis of such distal integration?

One possible mechanism explaining the activation of ganglion cells by distant stimuli would involve amacrine cells: they could propagate the activity of bipolar cells laterally to distant ganglion cells[15]. To test whether glycinergic amacrine cells are involved in the distant activation of ganglion cells, we blocked their synaptic transmission with strychnine (see Methods section). We found that the blocker suppressed the responses of distant cells to the randomly moving bar, while cells close to the bar still responded (Fig. 7a–c). As a result, under strychnine, distant subunit weights were zero.

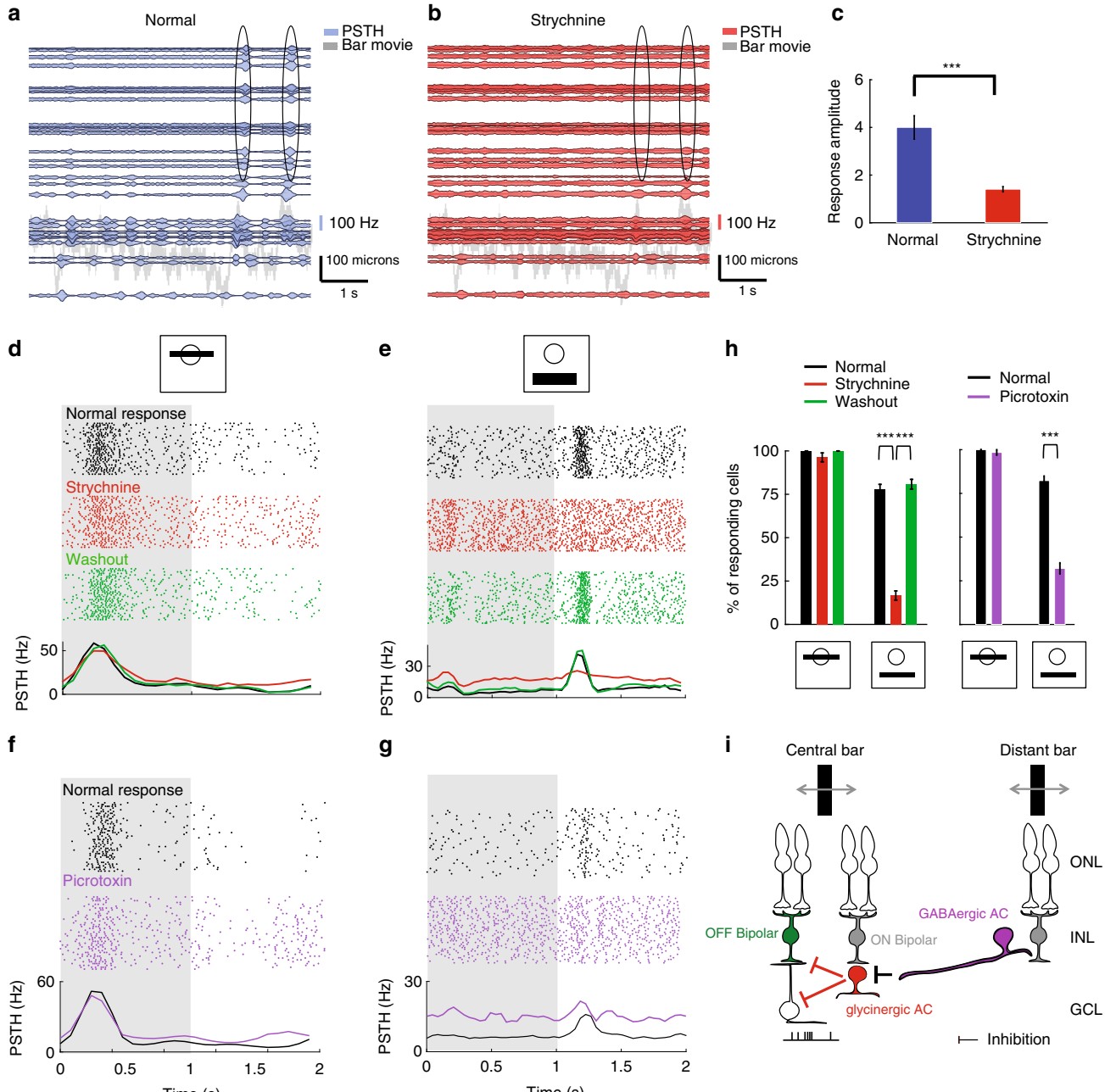

**Fig. 7** A disinhibitory circuit of amacrine cells is responsible for distant activation. **a** Responses of ganglion cells to a randomly moving bar, similar representation than Fig. 1c. Ellipse indicates responses of distant cells. **b** Same as **a** after application of strychnine. **c** Distant response amplitude (see Methods section) for normal (blue) and strychnine (red) condition ($n = 16$ cells). The three stars indicate that the p-value of a two-sample t-test was lower than 0.001. **d** Response to a bar flashed inside the receptive field center is not altered by the addition of strychnine. Single-cell responses to repeated presentations of the dark bar (shaded gray indicates the time window where the bar was presented); each dot is a spike, each row corresponds to a stimulus repeat. Black: control. Red: after addition of 1 µM strychnine to the bath. Green: after washing out strychnine. Bottom: PSTHs computed from the rasters. **e** Same as **a** for a bar flashed far from the receptive field center. **f** Single-cell responses to repeated presentations of the dark bar, same representation as in **d**. Black raster: control. Purple raster: after addition of 10 µM picrotoxin to the bath. Bottom: PSTHs computed from the rasters. **g** Same as **f** for a bar flashed far from the receptive field center. **h** Left histogram: percentage of cells responding to a central flash (left) and to a distant flash (right) before and after strychnine application, and after the drug wash-out. Same colors as **d**, **e**. Data are represented as mean ± SEM. Right histogram: same for picrotoxin. The three stars indicate that the p-value of a two-sample t-test was lower than 0.001. **i** Schematic showing a putative circuit for distant activation. When a distant stimulus is present, excitation of GABAergic amacrine cells (blue), through ON bipolar cells, inhibits subsequent glycinergic amacrine cells (red), which, in turn, disinhibits bipolar cells and ganglion cells. Similar glycinergic cells relay the output of ON bipolar cell inside the receptive field center to create a balance between ON and OFF inputs (see text). ONL: outer nuclear layer. INL: inner nuclear layer. GCL: ganglion cell layer

Strychnine also suppressed distant responses to a flashed bar (Fig. 7e, h), while leaving central responses mostly unaffected (Fig. 7d, h). Washing out the drug restored the distant responses (Fig. 7e, h), which confirms that glycinergic amacrine cells constitute a necessary component of the observed distant responses.

Central subunit weights were also affected by strychnine. We fitted our subunit model to the responses of the central cells and found that the blocker selectively suppressed the negative weights assigned to ON subunits in the receptive field center, and the negative weights assigned to OFF subunits in the near surround (Supplementary Fig. 8).

Suppression of distant responses by strychnine suggested that the weights assigned to distant subunits in our model are mediated by glycinergic amacrine cells. How could some of these weights be positive, while glycinergic amacrine cells have an inhibitory effect on their post-synaptic targets? One explanation is a disinhibitory loop (Fig. 7i), where one amacrine cell inhibits its post-synaptic amacrine cell target, which in turn disinhibits the ganglion cell (directly or through bipolar cells) (see also Manu and Baccus[43]). Such a disinhibitory circuit could involve serial connections between GABAergic and glycinergic amacrine cells[44], or, alternatively, serial connections between different types of glycinergic cells.

To tease apart these two hypotheses, we aimed at altering GABAergic transmission, and see if it could affect the distant responses. Because GABA blockers can trigger large oscillations across the retina after some time[40], they make it difficult to obtain a long and stable recording window. We therefore only tested how altering GABAergic transmission with picrotoxin affected distant responses to flashed bars (see Methods section). We found that picrotoxin decreased the responses to distant flashed bars, much more than it affected the responses to central bars (Fig. 7f–h). These results suggest that GABAergic cells might take part in this disinhibitory loop. Wide field GABAergic amacrine cells might play a key role to propagate the signal over large distances. An alternative hypothesis could be that gap junctions between amacrine cells help propagating the signal, but we did not find any significant decrease of the distant responses when trying to block gap junction with 18 $\beta$gA[12] (data not shown).

Our results thus suggest a disinhibitory loop of GABAergic and glycinergic amacrine cells that can ultimately inhibit OFF bipolar cells[45] or ganglion cells[46]. The net effect of such a disinhibitory circuit is a distant excitation of ganglion cells (Fig. 7i). The correspondence between subunit polarity and bipolar cell type is not guaranteed, but our results suggest that glycinergic amacrine cells might be involved in relaying to the recorded ganglion cells the activity of ON bipolar cells in the center, and the activity of OFF bipolar cells in the near surround (Supplementary Fig. 8). One possible disinhibitory circuit is thus that GABAergic amacrine cells inhibit glycinergic amacrine cells that receive input either from ON bipolar cells in the center, or from OFF bipolar cells located in the near surround. In both cases, these glycinergic amacrine cells will disinhibit ganglion cells, directly or through OFF bipolar terminals. These GABAergic amacrine cells are probably activated by ON or OFF bipolar cells. To determine the relative contribution of ON and OFF bipolar cells to this distant activation, we blocked the ON pathway with LAP-4 (see Methods section). We found that distant responses were mostly suppressed under the LAP-4 condition (Supplementary Fig. 9). The GABAergic amacrine cells are therefore mostly activated by ON bipolar cells, although we cannot exclude a smaller contribution from the OFF pathway.

## Discussion

We have shown that two representations of a stimulus co-exist, at the same time, within a neural population formed by ganglion cells of a single type. We constructed a mathematical model that recapitulated the multiplexing of the two relevant computations. To that end, the model required nonlinear summation within the receptive field, as well as a gain control mechanism. The model predicted precisely the responses of the fast OFF ganglion cells to a bank of dynamical stimuli which included complex, spatio-temporal stimulation in the far surround. Finally, our experiments suggested that a disinhibitory retinal circuit composed of two amacrine cells could mediate the distant computation.

When an object is moving randomly, neurons whose receptive field centers overlap with the object code for its position, while distant neurons code for general, large-scale changes in the stimulus (Fig. 8a, b). Each neuron can switch from one computation to the other depending on the visual context. Previous works have shown that the feature extracted by a cell can change when the average luminance changes[14,17], or during saccadic exploration of the visual scene[15]. Here we show that, in a single-visual scene, the same cell type can be used to extract two features simultaneously. Within this single type of cells, a subset of cells carries more information about position, and another, disjoint subset is not very sensitive to position. Feature extraction does not change only with the average luminance of the visual scene. Rather, two features can be extracted at the same time by a single cell type in a single-visual scene. These findings expand the traditional view of a "neural map" where there is a one-to-one correspondence between one cell type and one visual feature: here we show that a "neural map" can contain more than one "feature map" at the same time.

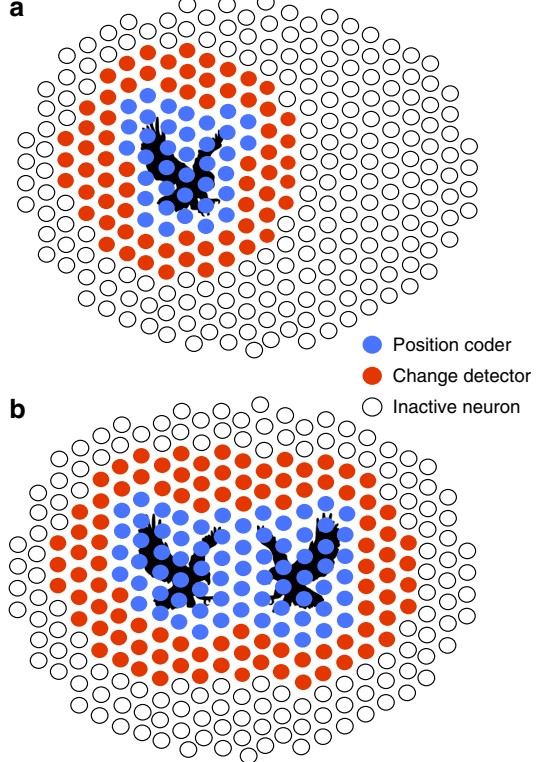

**Fig. 8** Cells of a single-type switch between two different computations. **a** Schematic of how cells of the fast OFF type code for a moving stimulus (here an eagle depicted in black). Each circle represents a fast OFF cell. Empty circles correspond to inactive cells, red circles to cells acting as change detectors, while blue cells are sensitive to the exact position of the stimulus. **b** Reorganization in the population code following the simultaneous display of two stimuli. Same legend as **a**

The idea that a single type of cells could extract two features from the visual scene has been suggested by several studies[47–49]. They demonstrated that the surrounds of ganglion cells can have different spatial scales or dynamics than the center, and this already questioned the idea that a single feature is extracted by a single type. However, it was unclear in which cases the surround would perform a nonlinear computation while the center performed a quasilinear computation, or if both center and surround carried out computations with similar degrees of nonlinearity. Here we have shown that the type of ganglion cells considered determines which computational scheme is implemented. In the type we focused on, surround performs a nonlinear computation that can be modeled with a two-layer network, while the center performs a quasilinear computation. In another type of ON cells, center and surround were both nonlinear. Our findings show that the multiplexing of computations can take multiple forms in different cell types. Our results suggest that the implementation of the quasilinear computation is performed by combining the inputs from OFF and ON bipolar cells in a "push–pull" manner: positive weights for the OFF subunit correspond to negative weights for the corresponding ON subunit. This push–pull organization linearizes the relation between the stimulus and the response. On the contrary, distant inputs from OFF and ON bipolar cells were not summed in a push–pull organization, and could not be linearized.

Multiplexing two computations in a single-neural type could enable optimal use of coding resources: if ganglion cells do not have an object inside their receptive field center, rather than staying silent, they are put to use to code for a different feature of the stimulus. However, having co-existent stimulus representations present in the same neural map may appear problematic for subsequent stages of processing: how can downstream neurons tease apart spikes corresponding to one or the other computation? Thanks to the simultaneous access to a large number of cells with the MEA technique, we could show that that a large ensemble of ganglion cells will respond synchronously and sparsely to distant objects. While such responses could appear negligible or ambiguous when observed at the single-cell level, access to the complete population would enable downstream processing to unambiguously recognize distant responses by detecting synchronous ganglion cell activity. Thus, synchrony could serve as a signature in the spike trains to separate position signals from change signals. A recent study has shown that single cell in the LGN can integrate over many ganglion cells[50]. It might be possible for some LGN cells to detect the synchronous activity of many ganglion cells. Our study suggests that two feature maps can be multiplexed in a single-cell type, but each map will be read out with a different kind of downstream neuron. On the one hand, a simple integration of the outputs from a few neurons should be able to isolate the linear, position-sensitive computation. On the other hand, a neuron acting as coincidence detector over a large population of ganglion cells should be able to detect the synchronous activation that is the signature of the nonlinear, position-insensitive computation. This multiplexing of computations in a single-cell type might therefore be a way to have several channels sending information to the brain, without compromising the stability of the visual representation, because each of these channels will be read out independently with a different downstream neuron. Since we only recorded from ganglion cells, the way ganglion cell activity is integrated remains to be tested, but it is worth noticing that recent studies show that different neurons in the LGN have very different ways to integrate ganglion cell activity, either from a few cells or from a large population[50].

Previous works have shown that ganglion cells can be activated by fast motion in their far surround ("shift-effect"[18,51–54]). Here we constructed a model that can accurately predict how fast OFF ganglion cells would respond to distant, complex stimuli, and how these distant stimuli would be integrated with other stimuli simultaneously displayed inside the receptive field center. Previous models mostly focused on how the surround modulates responses to central stimuli. However, how responses to distant stimuli can modulate ganglion cells themselves, and how they could be affected by center stimulation, has received less attention[36]. Hochstein and Shapley[47], Victor and Shapley[55] showed that Y ganglion cells could be understood with a subunit model similar to the one we used in Fig. 2d, together with a linear filter. While the responses to surround stimulation were clearly nonlinear, it was unclear if the responses to central stimulation were dominated by the linear or the nonlinear component of the model. This model explained the responses of a broad class of cells (ON and OFF Y cells), corresponding to multiple cell types, and our results show that some cell types have a dominant nonlinear component in the center, while the one we focused on here had a dominant linear component. The model proposed by Hochstein and Shapley[47], Victor and Shapley[55] is thus probably covering several types, corresponding to several ways of pooling subunit outputs in the center, either in a push–pull manner, so that the processing in the center is linearized, like in the OFF type we studied, or in a nonlinear manner, like in the ON type we showed in Fig. 3. Demb et al.[56] found that inputs from center and surround stimulation were summed linearly, while we found a nonlinear suppression of distant inputs. This discrepancy could be due to a difference of species, cell type, or recording technique. Passaglia et al.[20] showed that distant stimulation could be suppressed by center stimulation, but the timescale of the modulation was much longer than in our work. Interestingly, Jadzinsky and Baccus[57] suggested a model to predict how stimulation of the surround can affect the selectivity to the center stimulation that bears some similarity with our model. In most studies, the stimulus employed to modulate activity from the surround was very large. In our study, we showed that the same stimulus triggered two different types of responses, a central one and a distant one, within the same type of ganglion cell, demonstrating the coexistence of the two representations.

Our results suggest that the retinal network implements the activation of ganglion cells by distant stimuli through a disinhibitory circuit in which GABAergic amacrine cells are activated by bipolar cells and subsequently inhibit glycinergic amacrine cells. This release of glycinergic inhibition can affect both OFF bipolar cells[44] and OFF ganglion cells[46], and results in OFF ganglion cell activation. A similar disinhibitory circuit might also be involved in other kinds of complex processing taking place in the ganglion cell surround. When large visual features stimulate distant regions of the surround, the inhibitory input to bipolar cells[44] and ganglion cells[46] was reduced. This reduction of surround inhibition was mediated by a disinhibitory circuit similar to the one we uncovered. Elucidating the exact circuit that we hypothesized based on our pharmacology experiments would require targeting specific subtypes of amacrine cells genetically and perform intracellular recordings of these amacrine cells and ganglion cells simultaneously to demonstrate their implication in generating these distant responses. These challenging experiments are clearly beyond the scope of this study.

We have shown that a single-cell type mosaic can simultaneously multiplex several fundamentally distinct computations. Our findings considerably enrich the classical view of ganglion cell types as being tightly linked to their corresponding feature maps, and uncover the flexibility of the retinal code when stimulated with complex, dynamical stimuli. The notion of a feature map is central to most sensory structures. Flexible computations, where several features are represented by a cell type

simultaneously in response to complex stimuli, might also be implemented in other sensory areas. It remains to be understood whether this flexibility can be seen as arising from some efficient coding principle[58], and how such flexible coding schemes can be interpreted by the downstream areas[59].

## Methods

**Retinal recordings.** Recordings were performed on the Long-Evans adult rat of either sex. Animals were killed according to institutional animal care standards. The retina was isolated from the eye under dim illumination and transferred as quickly as possible into oxygenated Ames medium. The retina was then lowered with the ganglion cell side against a multi-electrode array whose electrodes were spaced by 60 μm, as previously described[25,26]. Raw voltage traces were digitized and stored for off-line analysis using a 252-channel preamplifier (MultiChannel Systems, Germany). The recordings were sorted using custom spike sorting software developed specifically for these arrays[25,26]. We extracted the activity of a total of 1098 neurons over 7 experiments with satisfying standard tests of stability and limited number of refractory period violations. No statistical method was used to pre-determine sample size. All relevant data and codes are available from the authors upon request.

**Visual stimulation.** Our stimulus was composed of one or two black bars moving randomly on a gray background. Each bar was animated by a Brownian motion, with additional feedback force to stay above the array, and repulsive forces so that they do not overlap. The bars stayed within an area that covers the whole recording array. The amplitude of the bar trajectories allowed them to sweep the whole recording zone. The trajectories of the bars $x_1$ and $x_2$ are described by the following equations[60]

$$\frac{dv_1}{dt} = -\frac{v_1}{\tau} + \text{sign}(x_1 - x_2)\left(\frac{R}{|x_1 - x_2|}\right)^6,$$
$$-\omega_0^2(x_1 - \mu_1) + \sigma W_1(t) \tag{1}$$

$$\frac{dv_2}{dt} = -\frac{v_2}{\tau} + \text{sign}(x_2 - x_1)\left(\frac{R}{|x_2 - x_1|}\right)^6,$$
$$-\omega_0^2(x_2 - \mu_2) + \sigma W_2(t) \tag{2}$$

where $W_1(t)$ and $W_2(t)$ are two Gaussian white noises of unit amplitude, $\mu_2 - \mu_1 = 600$ μm is the shift between the means, $\omega_0 = 1.04$ Hz, $\tau = 16.7$ ms, $R = 655$ μm and $\sigma = 21.2$ μm s$^{-3/2}$. The width of one bar is 100 μm and the length 2 mm. The stimulus was displayed using a Digital Mirror Device and focused on the photoreceptor plane using standard optics. For receptive field mapping, a random binary checkerboard was displayed for 1 h at 50 Hz (check size: 60 μm).

All the other stimuli used (for classification of cells, fitting the gain control model and pharmacological study) are described in the corresponding method section. For all stimuli, the level of light of the gray background was between $10^{12}$ and $10^{13}$ photons cm$^{-2}$ s$^{-1}$. In Supplementary Fig. 2 we varied this level from $10^{10}$ to $10^{14}$ photons cm$^{-2}$ s$^{-1}$.

**Moving textures.** Each texture consisted of an alternation of white and black stripes of width 20 μm, chosen randomly with equal probability. The two textures were displayed next to each other (aligned on the axis perpendicular to the stripes) and were animated by independent Brownian motions of the form:

$$\frac{dv_1}{dt} = -\frac{v_1}{\tau} + \sigma W_1(t), \tag{3}$$

$$\frac{dv_2}{dt} = -\frac{v_2}{\tau} + \sigma W_2(t), \tag{4}$$

where $W_1(t)$ and $W_2(t)$ are two Gaussian white noises of unit amplitude and $\sigma = 21.2$ μm s$^{-3/2}$. In order to confine each texture to a given location of the screen, we used a rectangular mask of dimension $1700 \times 870$ μm$^2$, with the big axis of the rectangle corresponding to the direction of the stripes. The screen was white outside of the mask when only one texture was projected.

**Typing.** We performed cell classification based on the response of the cells to a set of stimuli and on their temporal receptive field.

*Full field flicker*: this stimulus consisted of a 15-s sequence of a full-field stimulus, repeated 100 times. The stimulus was generated by selecting a random row of pixels from a natural image and displaying subsequently at 40 Hz the intensity of these pixels uniformly on the entire screen.

*Shifting barcode*: this stimulus consisted of an alternation of white and black stripes of width 70 μm chosen randomly, moving at a constant speed of 1000 μm s$^{-1}$ in the 4 cardinal directions. For each direction, the 17-s sequence was repeated 30 times.

For each cell, we created a vector by concatenating the PSTH in response to the *full field flicker* stimulus, the 4 PSTHs in response to the *shifting barcode* stimulus

corresponding to the 4 cardinal directions, the temporal receptive field and the auto-correlogram of the cell in response to the checkerboard stimulus. The PSTHs of the *shifting barcode* were temporally realigned beforehand according to the receptive field location of each cell. PSTH for each stimulus was normalized such that they all had a mean of 0 and a variance of 1.

We then performed PCA on this collection of vectors. We kept the projections on the first eigenvectors in order to explain 95% of the total variance. We then performed clustering on these vectors using the peak density algorithm[61]. The threshold parameters of the algorithm were manually adjusted in order to select the outliers as centroids of the clusters. This method allowed us to identify reliably an OFF type of ganglion cells across all experiments. The receptive fields (RF) were regularly tiling the visual field, with little overlap between them. This mosaic property, often observed in the retina, was used here as a validation of our typing procedure, as we did not use the position of the RFs in the clustering procedure.

**Synchrony between cells.** To quantify the synchrony between cells, we displayed a 10-s bar movie to the retina, repeated 54 times. A maximum of 25 cells of the same type recorded simultaneously were subdivided in two groups, the distant cells, that were more than 200 μm away from the central bar position, and the central cells, that were less than 200 μm away from the central bar position. For all cells we computed the PSTH with a time bin of 20 ms. We computed the Pearson coefficient between all pairs of PSTHs of distant cells, and all pairs of PSTHs of central cells, respectively. We grouped the pairs based on the distance between their receptive field centers along the bar motion axis.

**Quasilinear model (LN) and subunit model (LN-LN).** The subunit model is a two-layer model that predicts the response of a ganglion cell to the moving bar. Each layer performs a linear combination of its inputs followed by a nonlinear transformation. The first layer is a collection of identical and translated Linear–NonLinear (LN) units. The second layer is a unique LN unit taking the output of the first layer as an input.

In the first layer, we tiled the space with 100 bipolar-like OFF subunits, and 100 ON subunits, on a one-dimensional lattice, with subunits equally spaced at 20 μm interval. Each unit had a receptive field with a Gaussian spatial profile of the right polarity and a biphasic temporal profile, modeled by a sinusoid. All units of a same polarity are identical up to a translation. The nonlinearity was a rectified square function, $h$. The output of the ON subunit layer was therefore:

$$F(x,t) = h\Big(\int_{t-T_{\text{subunit}}}^{t}$$
$$\int_{-\infty}^{\infty} \sin(2\pi\omega(t-t'))e^{-\frac{(x-x')^2}{2\sigma^2}}s(x',t')dx'dt'\Big), \tag{5}$$

where $h(x) = x^2$ if $x \geq 0$, and 0 otherwise. For the OFF subunit, the equation was the same as ON subunit, except that was multiplied by $-1$. $T_{\text{subunit}} = 0.3$ s, $\omega = 1/T_{\text{subunit}}$, $\sigma = 30$ μm.

The stimulus movie $s(x,t)$ was one-dimensional in space because the stimulus was a long bar, whose length can be considered infinite. We used a temporal binning of 17 ms, corresponding to the refresh rate (60 Hz) of the screen used to project the movie on the retina.

The second layer consisted of a single Linear–NonLinear Poisson unit. The unit pooled linearly its inputs from all the subunits of the first layer according to a kernel $K$, with an extension in time of 0.5 s. To obtain the firing rate $r(t)$ of the cell, the weighted sum was passed through a nonlinearity of the form $f(x) = \log(1 + \exp(x))$. The spikes were then generated according to a Poisson process. The firing rate is therefore equal to:

$$r(t) = \alpha \log(1 + \exp(\beta G(t) + \theta)), \tag{6}$$

where

$$G(t) = \int_0^{T_{\text{filter}}} \int_x K(x,t')F(x,t-t')dxdt', \tag{7}$$

with $T_{\text{filter}} = 0.5$ s, and $\alpha$, $\beta$, $\theta$ are parameters of the nonlinearity that are fitted to the data.

The quasilinear model (LN) was built using the same architecture as the subunit model, except that the rectified square nonlinearities in the subunits were replaced by the identity. We obtained the following values for the parameters $\alpha$, $\beta$, $\theta$: for the LN model, $\alpha = 8.2 \pm 7.1$, $\beta = 3.9 \pm 0.4$, $\theta = -0.8 \pm 0.1$. For the subunit model, $\alpha = 9.6 \pm 1.3$, $\beta = 3.11 \pm 0.6$, $\theta = -2.7 \pm 0.5$. We tried to replace the last nonlinearity with a parametrized cubic spline nonlinearity following[62], but this did not improve the performance of the model (data not shown).

**Model fitting.** For both models, we used the same fitting procedure. The parameters of the kernel $K$ and the parameters of the spiking nonlinearity $\alpha$, $\beta$, $\theta$ were the only parameters fitted to the data. The kernel parameters and the spiking nonlinearity parameters were fitted alternatively using block gradient descent[32] across 6 iterations. The repeated parts of the stimulus were held back during fitting and were used to cross-validate the model.

The parameters of the kernel were optimized to maximize the log-likelihood function of the spike train under Poisson assumption[32]. For this optimization we performed Limited-memory Broyden–Fletcher–Goldfarb–Shanno gradient descent on the parameters of the kernel[32]. In order to avoid overfitting, we imposed two regularization constraints: spatio-temporal smoothness and sparseness of the kernel. The cost function $C$ was of the form:

$$C = \sum_t -LL(s_{obs}(t)|s_{pred}(t)) + \lambda_{XT}\|L_{Kron}K\|_2 + \lambda_1\|K\|_1, \quad (8)$$

where $LL$ is the log-likelihood of the observed spike train $s_{obs}$ (under Poisson assumption), $K$ is the kernel defined above, $\lambda_{XT} = 300$ is the penalty term enforcing smoothness of the kernel, $L_{Kron}$ is the Kronecker sum of discrete Laplacians, $\lambda_1 = 400$ is the L1 penalty term enforcing sparseness of the filter coefficients.

The penalty terms were chosen to minimize overfitting. To fit the linear model (LN), we divided by 10 these two penalty terms as it slightly improved the performance of the model for distant cells. The parameters of the nonlinearity were fitted by minimizing the cost function with the active-set method. The following constraints were enforced: $\alpha > 0$, $\beta > 0$, $\theta$ has an upper bound. $\beta$ and $\theta$ were redundant with the kernel parameters but adding them accelerated the convergence of the optimization[32]. The fitting was performed for each cell independently.

**Performance quantification of LN and subunit models**. We fitted the model on the unrepeated part of the stimulus and we tested the performance of the model on the repeated part of the stimulus (54 repetitions of a 10 s sequence). For each cell we then computed the Pearson coefficient $r$ between the real PSTH and the predicted PSTH (time bin: 17 ms). Population averages are indicated in the text as mean ± standard error of the mean.

In order to show that the LN model was performing significantly better for central stimulation than for distant stimulation, we selected only the cells that were less than 300 μm away from the bar in one condition and more than 400 μm away from the bar in the other condition. We then performed a paired $t$-test comparing the performance of the LN model in both conditions for each cell.

A possible explanation for why the linear model performed poorly for distant cells could be that distant stimulation evoked less reliable responses. In order to exclude this possibility, we computed the ratio of explainable variability predicted by the model. The explainable variability was defined as the average Pearson coefficient between pairs of PSTHs generated by instantiations of a Poisson process with mean firing rate equal to the real firing rate of the cell estimated from the PSTH. We divided the performance of our model (defined as the Pearson coefficient between real and predicted PSTH) by this explainable variability to obtain the ratio of explainable variability predicted by our model.

**Calculation of average linear filters from the subunit model**. To compute the average filter in the one-bar condition (Fig. 4d, f), we selected only the cells stimulated outside of their receptive field (RF) center. Our criterion was that the bar central position should be more than 200 μm away from the RF center.

To compute the average filter in the two-bar condition (Fig. 4c, e), we selected only the cells that were stimulated inside their receptive field centers by at least one of the bars. Our criterion was that the bar central position should be less than 200 μm away from the RF center.

For all cells and in both bar conditions, only a portion of the extended receptive field center was visited by the bars, therefore inducing a bias in the filters fitted on these movies. To compute the weights of the average filter without bias, we first realigned the filter of each cell relative to the center of its receptive field. Then for each coordinate $(x, t)$ of the average filter we averaged the corresponding subunit weights for the subset of cells for which the coordinate was visited more than 200 times per hour by the bar. Note that this implies that the subunit weights for the center are not well estimated since they are not visited often by the distant bar.

**Subunit and LN model for the texture stimulus**. As for the bar movie, the subunit model for the texture is a two-layer model that predicts the response of a ganglion cell to the moving texture. Each layer performs a linear combination of its inputs followed by a nonlinear transformation. The first layer is a collection of identical and translated Linear–NonLinear (LN) units. The second layer is a unique LN unit taking the output of the first layer as an input.

In the first layer, we tiled the space with respectively 34 bipolar-like ON and OFF subunits on a one-dimensional lattice, with subunits equally spaced at 20 μm interval. Each unit had a receptive field with a Gaussian spatial profile of the right polarity and a biphasic temporal profile, modeled by a sinusoid applied to a power transformation of time $g$, so that the second lobe was more extended in time that the first lobe. All units of a same polarity are identical up to a translation. The nonlinearity was a rectified power 4 function, $h$. The output of the first layer was therefore:

$$F(x, t) = h\left(\int_{t-T_{subunit}}^{t}\int_{-\infty}^{\infty} N(\sin(2\pi\omega g(t-t')))e^{-\frac{(x-x')^2}{2\sigma^2}}s(x', t')dx'dt'\right), \quad (9)$$

where $h(x) = x^4$ if $x \geq 0$, and 0 otherwise, $g(t-t') = (t-t')^5$, $N$ is a normalizing function ensuring that the positive lobe and the negative lobe of the modified sinusoid function

both have a L1 norm equal to 1 (same area under the curve), and $T_{subunit} = 1$ s, $\omega = 1/T_{subunit}$, $\sigma = 30$ μm

As for the bar movie, the stimulus movie $s(x, t)$ was one-dimensional in space. We used a temporal binning of 17 ms, corresponding to the refresh rate (60 Hz) of the screen used to project the movie on the retina.

As for the bar movie, the second layer consisted of a single Linear–NonLinear Poisson unit. The unit pooled linearly its inputs from all the subunits of the first layer according to a kernel $K$, with an extension in time of 0.13 s. To obtain the firing rate $r(t)$ of the cell, the weighted sum was passed through a nonlinearity of the form $f(x) = \log(1 + \exp(x))$. The spikes were then generated according to a Poisson process.

$$r(t) = \alpha\log(1 + \exp(\beta G(t) + \theta)), \quad (10)$$

where

$$G(t) = \int_{T1}^{T2}\int_x K(x, t')F(x, t-t')dxdt', \quad (11)$$

with $T1 = 0.13$ s and $T2 = 0.63$ s, and $\alpha, \beta, \theta$ are parameters of the nonlinearity that are fitted to the data.

The quasilinear model (LN) was built using the same architecture as the subunit model, except that the rectified power 4 nonlinearities in the subunits were replaced by the identity, and $T1 = 0$ s, $T2 = 0.5$ s. We obtained the following values for the parameters $\alpha, \beta, \theta$: for the LN model, $\alpha = 0.4 \pm 0.3$, $\beta = 6.2 \pm 0.6$, $\theta = -0.2 \pm 0.2$. For the subunit model, $\alpha = 0.2 \pm 0.02$, $\beta = 0.8 \pm 0.35$, $\theta = -1.4 \pm 0.8$.

**Suppression index**. In Fig. 4b, we quantified the suppression of the response to the distant bar when there was another bar moving inside its receptive field center. For this we defined the residual response to the distant bar in case of a central bar as:

$$PSTH_{residual} = PSTH_{both\,bars} - PSTH_{central\,bar}. \quad (12)$$

We then computed the suppression index, defined as:

$$I_{supp} = 1 - r(PSTH_{distant\,bar}, PSTH_{residual}), \quad (13)$$

where $r$ is the Pearson coefficient. If the suppression of the distant bar response is complete, the index should be equal to one. If there is no suppression, and the responses to each bar are summed, then the index should be equal to 0 in the absence of noise. However, since noise is present, we defined a suppression index due to noise, which reflects the index value that should be expected purely from noise, without suppression of the distant response:

$$I_{supp}^{noise} = 1 - r(PSTH_{distant\,bar}^1, PSTH_{distant\,bar}^2), \quad (14)$$

where $PSTH_{distant\,bar}^1$ and $PSTH_{distant\,bar}^2$ were computed on two different sets of trials. We performed this quantification on the 25 cells recorded and plotted the mean and SEM of the suppression index for the real data and the one due to noise. A suppression index higher than $I_{supp}^{linear}$ indicates a true suppression that cannot be explained by noise.

**Gain control model**. We displayed two bars of width 300 μm and separated by 800 μm, oscillating with a sine wave trajectory of amplitude 350 μm at slightly different frequencies: the central bar was oscillating at 2 Hz and the distant bar at 1.98 Hz. The central bar was played at 8 different contrasts interleaved randomly and ranging linearly from 0 to 1. For each contrast, the two bars were oscillating during an uninterrupted sequence of 50 s, so that the central bar had traveled exactly 100 periods and the distant bar exactly 99 periods during a sequence. At the end of a sequence, all possible phase shifts between the two bars had been visited exactly once. This trick allowed us to average out the influence of one bar when computing the PSTH on the period of the other bar.

To show the gradual suppression of the distant response in Fig. 5b, we normalized the amplitude of the response to the distant bar by the amplitude of the response to the distant bar alone (i.e., zero contrast for the central bar).

We then fitted a single model on all contrast conditions. The model was of the form:

$$R_{gain}(t) = \frac{R(t)}{1 + H\int_{t-\tau}^t R(t')dt'}, \quad (15)$$

where $\tau = 1$ s is the time constant of integration of the gain control and $H$ is the gain. $R(t)$ is the total response before application of the gain control, given by the equation:

$$R(t) = \alpha_0 \cdot c \cdot r_c(t) + \alpha_1(c \cdot r_c(t))^2 + \beta_0 \cdot r_d(t) + \beta_1 \cdot r_d(t)^2, \quad (16)$$

where $r_c$ is the response to the central bar alone at full contrast, $r_d$ the response to the distant bar alone, $c$ is the contrast. $r_c$ and $r_d$ were estimated from the PSTHs in response to the central bar and to the distant bar played alone, respectively. We

needed to introduce quadratic terms because the PSTH for the central bar condition depended quadratically on the contrast of the central bar. This is consistent with our subunit model, where the first layer contained a rectified quadratic function $h$.

We fitted the parameters $\alpha_0$, $\alpha_1$, $\beta_0$, $\beta_1$, and $H$ so as to maximize the log-likelihood of the spike train under Poisson assumption (bin size: 17 ms). To adjust the parameters, we used the active-set method. However, we fixed the parameter $\tau$ to 1 s because the periodicity of the stimulus did not allow us to explore thoroughly the time constant of integration of the gain. We obtained the following values for the parameters (average and SEM): $\alpha_0 = 7.4 \pm 0.8$, $\alpha_1 = 31 \pm 4$, $\beta_0 = 2.3 \pm 0.3$, $\beta_1 = 11.6 \pm 1.5$ and $H = 8.6 \pm 0.4$. To test our model, we measured for each cell ($n = 21$) and each contrast the amplitude of the response to the distant bar (defined as max (PSTH)-min(PSTH), bin: 100 ms) and compared it to the amplitude predicted by our model. We then estimated the percentage of variance explained by our model across all cells and conditions using bootstrapping.

Fitting the gain control model is challenging because this is a non-convex optimization. For randomly moving bar stimuli, an additional challenge is the sparsity of responses to the distant stimulus, while responses to the central stimulus are much denser: when trying to fit the model to these responses, the optimization will give more weight to the denser responses. As a result, we did not manage to fit the gain control model to randomly moving bars, and we used oscillating bars, where responses to central and distant stimulations can be separated. However, the oscillating bar stimulus does not allow fitting the subunit model. We would therefore need to record the same cells with both the randomly moving bar stimuli to learn the subunit model, and the oscillating bar stimulus to learn the gain control model. Unfortunately, it is impossible to obtain a stable recording that can allow displaying the two stimuli. We therefore used the PSTHs as a proxy for the output of the subunit model and quasilinear model, instead of the filtered subunit outputs.

**Information estimation.** The information conveyed by the cell response $R$ about the stimulus $X$ (i.e., mutual information between $R$ and X) is equal to the reduction in entropy of the distribution of $X$ provided by the knowledge of $R$. It was calculated directly on ganglion cell responses, not using the models described above.

$$I(R, X) = H(X) - H(X|R). \qquad (17)$$

In our case, we first defined the stimulus as the position $P(t + \delta t)$ of the moving bar for different lags $\delta t$ relative to the cell response $R(t)$ (in Fig. 6f, $\delta t$ is the x-axis of the insets). The lags were introduced to account for the delay in the neural response. We discretized linearly the space of P in 10 bins in order to have a well-sampled distribution with our finite data set. We discretized the spike train in 10 ms bins and we binarized it by setting to 1 all the bins where there was at least one spike and to 0 the other bins. Changing the discretization steps used to bin P and the spike train did not change qualitatively our results. Then we computed the mutual information between the cell response and the instantaneous position of the bar with a lag $\delta t$ ranging from $-1$ s (information about the past stimulus) to 1 s (information about the future stimulus):

$$
\begin{aligned}
I(R(t), P(t + \delta t)) = H(P(t)) & \\
- [p(R(t) = 1)H(P(t + \delta t)|R(t) = 1) & . \qquad (18) \\
+ p(R(t) = 0)H(P(t + \delta t)|R(t) = 0)] &
\end{aligned}
$$

Note that the information about the future of the stimulus was not always zero. This is because the successive positions of the bar are correlated in time, so that part of the information conveyed by the cell response about the past position of the bar is also informative about the future position of the bar. We then defined the stimulus as the speed of the bar S with different lags $\delta t$ relative to the cell response. The speed was defined as:

$$S(t) = \frac{|P(t) - P(t - \tau)|}{\tau}, \qquad (19)$$

where $\tau = 100$ ms. We discretized linearly the space of S in 10 bins and we computed mutual information between $R(t)$ and $S(t + \delta t)$. To estimate the information rate in the insets of Fig. 6f, we divided the mutual information by the bin size (10 ms). For each cell, we finally computed the ratio between the maximum of $I(R(t), S(t + \delta t))$ and the maximum of $I(R(t), P(t + \delta t))$ over all time lags tested.

**Pharmacology.** To block glycinergic transmission, we added 1 µM strychnine (Sigma-Aldrich ref. S8753) to the bath[63–66]. To generate the rasters and PSTHs in response to the central bar, we flashed a dark bar of width 100 µm in the center of the receptive field of the cell for 0.5 s 40 times, separated by 0.5 s of gray screen. For the distant responses, we used 230 µm wide bars flashed for 1 s, in a region 0.5–1 mm away of the cell's receptive field center.

To reduce GABA transmission, we added 10 µM of the $GABA_A$ receptor antagonist picrotoxin (Sigma-Aldrich ref. P1675-5G) to the bath[67,68]. Note that $GABA_C$ receptors in rat are not effectively blocked by picrotoxin[69,70], unlike many other species. Usually, picrotoxin is used at a concentration of 100 µM or more in the mouse and rat retinas[16,71–73]. However, at that concentration, the retina

entered in an oscillatory regime immediately. By reducing the concentration of picrotoxin by a 10-fold factor, we could alter the GABA transmission while remaining closer to a natural regime. As a result, in many cases, the response to distant bar was decreased but a small response remained.

To block the ON bipolar pathway, we added 10 µM of the group III metabotropic glutamate receptors L-2-amino-4-phosphonobutyric acid (LAP-4) to the bath. We checked that ON responses to a flash were blocked. We then displayed a repeated sequence of a randomly moving bar and quantified the response modulation of central and distant cells.

For the population analysis, we flashed a bar 100 µm wide in random locations relative to the receptive fields of the cells, 20 times at each location. For each cell recorded of the type under study (19 cells for picrotoxin, 17 cells for strychnine), we selected the flashes that were less than 80 µm away from the receptive field center to study the effect of central stimulation. To study the effect of distant stimulation, we selected the flashes that were between 200 and 500 µm away from the cell receptive field center. For each stimulus and each cell, significant responses were determined based on a $z$-score analysis. We estimated the mean and standard deviation (SD) of the activity prior to stimulus and considered that a response was detected if the activity exceeded the mean by more than five times the SD in the second following the onset of the stimulus (for a bin size of 40 ms). To estimate the percentage of responding cells in Fig. 7h, we estimated means and standard errors of mean by pooling together all stimulus conditions across all the cells. We performed a one-tailed two-sample $t$-test to assess the reduction of responses to the distant flash after drug was added to the bath. For both picrotoxin and strychnine, the $p$-value was $<10^{-3}$.

To analyze the response to a randomly moving bar under strychnine and LAP-4, we estimated response modulation from the PSTHs of the responses as $\frac{\text{Max(PSTH)} - \text{Min(PSTH)}}{m}$, where $m$ was the average firing rate. $p$-value was $<10^{-3}$ for strychnine, and $<0.05$ for LAP-4.

**Error bars.** Unless stated otherwise, all error bars in figures and text are standard error of the mean (SEM). SD stands for standard deviation.

**Data availability.** All relevant data are available from the authors upon request.

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

## Acknowledgements

We thank Vicente Botella-Soller, David Kastner, Stuart Trenholm and Michael J. Berry II for fruitful discussions, Valérie Fradot for excellent technical support, Hanen Khabou for her illustrations, and Thierry Mora for critical reading of the manuscript. This work was supported by ANR-14-CE13-0003 to P.Y., ANR TRAJECTORY, ANR OPTIMA, the French State program Investissements d'Avenir managed by the Agence Nationale de la Recherche [LIFESENSES: ANR-10-LABX-65], a EC grant from the Human Brain Project (FP7-604102)), NIH grant U01NS090501 and AVIESAN-UNADEV grant to OM, a Foundation Fighting Blindness grant to S.P., and the Austrian Research Foundation FWF P25651 to GT. S.D. was supported by a PhD fellowship "DIM cerveau et pensee" from the region Ile-de-France.

## Author contributions

S.D., S.P., G.T., and O.M. conceived the study. S.D., E.M, R.C., and O.M. performed the experiments. S.D, U.F., P.Y., G.T., and O.M. analyzed the data. S.D., U.F., E.M., P.Y., G.T., and O.M. wrote the paper.

## Additional information

**Competing interests:** The authors declare no competing financial interests.

