## [Peer Review File · Nature Communications]

Reviewers' comments:

Reviewer #1 (Remarks to the Author):

Deny et al. combined experimental and computational modeling approaches to show that one type of retinal ganglion cells responds differently - and thus performs fundamentally different computations - when a light stimulus is presented inside or outside the receptive field center. The manuscript is written logically and the overall claim is supported by the results. As detailed below, however, I think that some critical data are missing to fully support it. The authors will then benefit a lot from further analyses and clarification.

=== Major points ===

(1) Linear versus nonlinear computations:

LN model is not "linear". The parameters enter linearly in the model (through the "static-nonlinear" link function), but the predictors (output responses) are nonlinear in relation to the input stimulus. It needs to be clarified what "computation" the authors are referring to. Or, it might be fair to say that the computation is "linearizable", rather than linear.

line 97: show the pooling weights for the LN model in the figure. In general, please report all the parameter values in each model for each condition.

line 98: the subunit models have quadratic rectifications. Summing up the outputs of ON and OFF subunits was then not linear, even though they have opposite response polarity and thus can contribute to "linearize" the inputs to a ganglion cell by cross-over inhibition (ie, "push-pull" circuits).

(2) ON versus OFF subunit pathways:

The authors should analyze not just the ON subunit circuits but also the OFF subunit circuits. Off subunits have an order of magnitude stronger effects than On subunits (Fig 3), and thus I don't think it is reasonable to focus only on the On pathway. Moreover, Off subunit weights for the distant stimulus change the sign under the two conditions (supp fig 1): negative values in combined bar conditions, but positive values in single bar condition. (Note, in contrast, On subunit weights have antagonistic center-surround structure in both conditions.) This change in the Off subunit weights seems to have a stronger effect than that in the On subunit weights (I cannot judge from the hue plots in fig 3 and supp fig 1), and it seems to me that this is the key difference between the center and surround computations.

Figure 3: show the weights for the OFF subunits here (but not as a supplemental). Also, show the weights (or summary statistic) for ON and OFF subunits in the same scale under the two conditions. This may be done better with line plots or bar graphs, but not with a hue.

Also, the gain control model should use the output of the subunit models, but not the data PSTHs, to examine how the gain changes for individual subunits. One can then analyze the gain control on the ON and OFF pathways separately. Similarly, data from pharmacology experiments should also be analyzed by using the subunit models to elaborate the ON and OFF circuits separately.

(3) Disinhibitory circuit:

Experimental data are too weak to fully support disinhibitory circuits. All the authors showed was that antagonistic surround response to a flashing bar (why not using the randomly moving bars?) is suppressed by strychnine. Disinhibitory circuits may work for ON subunit pathway, but not for OFF.

How is each of the ON and OFF pathways affected?

Also, what is the difference between the "classic" antagonistic surround responses and the distant responses the authors are reporting here? Classic explanation of antagonistic surround responses is through lateral inhibition via horizontal cells. Is there any contribution of such circuits? Also, is there any contribution of gap junctions? The pharmacology experiments with randomly moving bars, instead of flashing bars, will be more suitable to compare the changes of the parameter values in the subunit model.

Glycinergic amacrine cells have a narrow field. Wide-field GABAergic amacrine cells are most likely required if indeed disinhibitory circuits underlie the reported phenomena. But, if push-pull mechanisms are involved in the center response (to "linearize" the synaptic inputs to ganglion cells), why there is no change in either the evoked or baseline firing after applying strychnine (fig 6A,C)? Narrow-field glycinergic amacrine cells are most likely involved here, and I expect that strychnine affect these cells.

(4) Methods clarification:

Details are missing from the methods section. It is critical to describe everything in the methods; otherwise no one else can reproduce the results.

Typing: Need more details. How many types did the authors obtain? How did the authors classify kinetics (FAST vs SLOW) within OFF type cells? Also, among the previously reported cell types, which is it that (most likely) correspond to the one reported here?

Subunit models: are there 100 subunits each for ON and OFF subunits (line 385)? The Eq.(3) seems to define only OFF subunits, but what is the definition of the ON subunits? In particular, do the authors use different parameters between ON and OFF subunits, for example, to take into account that ON responses are generally slower than OFF responses? Also report the obtained parameters values (alpha, beta, theta) after fitting.

Fitting: is fitting done for each cell? I assume that the reported weights are obtained from population data (because moving bars jitter only a limited range), but would it be beneficial to accumulate population data first and perform data fitting (rather than doing data fitting first and accumulate the parameters to analyze the weights as in figure 3)?

Gain control model: the authors should use the subunit model outputs for r_c and r_s in Eq.(11), instead of the data PSTHs, especially for the r_s (see above). In particular, if the authors claim that the subunit model should be used to explain the distant stimulus response, then would it make more sense to examine the gain change for the output of those subunits separately? Also, report the fitted parameter values (α_0 , α_1 , β_0 , β_1 , and H). Also, any idea how the gain control is implemented?

=== Minor points ===

(1) Page 1, last line: better to say stimulus "location".

(2) line 30: Say "Individual cells switched from this computation", instead of "one computation"

(3) lines 79-80. This is going too far. The LN works well for a randomly moving stimulus, but it was not shown that it works well for a general stimulus, say, white-noise, natural stimulus, etc.

- (4) Figure 1: How about the distant cells on the other side (Fig 1E)? Are cells on the opposite sides also responding synchronously?
- (5) Figure 1D: Show all data points from individual pairs. How about central-distance cell pairs? Also, how about correlation at the spike level (finer time resolution) instead of PSTH?
- (6) Line 170. "The subunit model equipped with gain control" This is not directly tested. The methods section says that the gain control model uses data PSTHs, but not the ones from subunit models.
- (7) Line 171. Response to "randomly moving bar" was not tested. According to the methods section, it was sinusoidal waves that the authors used to test the gain control model.
- (8) Figure 4: how does the random gain control model work for randomly moving bar stimuli?
- (9) line 198: Change "Fig 3" to "Fig 3C"
- (10) line 203: remove comma
- (11) line 337: what is the bar length?
- (12) line 417: spell out BFGS (Broyden-Fletcher-Goldfarb-Shanno)
- (13) line 479: what is the bar length? Also, what is the amplitude of the oscillating sign wave?
- (14) Eq.(10): integral should be from $t-\tau$ to t (remove "prime" on the top t).
- (15) line 497: show data to support the claim (quadratic dependence of the PSTH).
- (16) Eq.(2) in supplement: use three dots to represent the range: $\tau=T...T+DT+L$
- (17) line 31 in supplement: should read "same property as the one found on"

Reviewer #2 (Remarks to the Author):

In their study, Deny et al. experimentally characterize and model the receptive field organization of a transient OFF-RGC type (OFFT-RGCs) in rat retina, and trace the influence of this receptive field organization on population coding of motion information. In particular, the authors show that receptive field center and surround of these cells are driven by nonlinear subunits, predominantly OFF-signed in the center and ON-signed in the surround. Due in part to nonlinear spatial integration, OFFT-RGCs respond to motion in both the center and the surround. Interestingly, the authors show that responses to surround motion are suppressed by center stimulation, which they explain by a global gain control mechanisms implemented in a cascade model after spatial summation. Deny et al. further show that responses to motion in the receptive field center encode the position of a moving object, whereas responses to distant motion encode the speed of a moving object. This implies that at any particular time different OFFT-RGCs in a population encode distinct features of motion. Finally the authors provide pharmacological evidence that responses to movement of distant stimuli are mediated by a disinhibitory circuit involving glycinergic amacrine cells.

Overall this is an interesting study. The experiments appear carefully conducted, and their motivation

and results are explained clearly in the text and illustrated well by the figures. It is in many ways quite a traditional multielectrode exploration of visual processing in the retina, and shares some of the weaknesses of this approach. By the standards of modern neuroscience, the understanding of circuit mechanisms underlying particular responses remains vague, as recordings are limited to spikes of the output neuron of the circuit. The fact that responses can be described by a model is orthogonal to experiments that would e.g. record from presynaptic neurons to test specific mechanisms of computation. Moreover, while one can show that different parts of a neuronal population can encode different parameters of motion (position vs. speed), there is no evidence that / how the brain uses this information. These criticisms are not unique to the present study, and the authors do not overstate or -interpret their findings. I mention them nonetheless, because given the inherent limitations of the approach, it is particularly important that the authors provide clarify the robustness, specificity, and context of their findings to warrant publication in a top-tier journal.

Specific comments

- 1) It is important to clarify to what extent responses to center and distant motion, and the suppressive effects of center motion on distant stimuli are specific to OFFT-RGCs or are observed in other ON, OFF and ON-OFF RGCs. The authors should include data from other cell types to illustrate differences or similarities in their motion responses and receptive field architectures.
- 2) How robust is the described receptive field architecture and motion encoding to stimulus conditions. The authors should include data from scotopic stimulation in addition to their current photopic data. This would help clarify whether motion information available to the brain via OFFT-RGCs is similar or qualitatively different depending on ambient illumination. There is some indication in the literature that transient OFF RGCs in rodents switch responses as a function of luminance (Farrow et al. Neuron 2013).
- 3) The receptive field model that the authors present accurately predicts responses to stimuli of the sort used in its construction. It is important for the authors to show whether their model generalizes e.g. to predict motion of textures or patterns rather than bars.
- 4) Regarding context. Transient OFF RGCs have frequently been characterized in mice and have been identified as a particular molecular and morphological type in mice (Huberman et al. Neuron 2008, Munch et al. Nat Neurosci 2009, Farrow et al. 2013, Murphy et al Neuron 2006...). The authors should attempt to establish whether the cells they record in rats are a homologous type by comparing characteristic response parameters.

Reviewer #3 (Remarks to the Author):

Multiplexed computations in retinal ganglion cells of a single type

The authors record and analyze large-scale multineuronal spiking activity in the ganglion cell layer of the rat retina. They interpret their findings as showing that a single cell type (fast OFF ganglion cells) codes in two different ways: (i) a quasilinear mode that dominates for objects that are near the receptive field center and represents position, and (ii) a nonlinear mode that dominates for objects that are far from the receptive field center and primarily represents speed.

There are several findings and contributions reported in this paper, and, because of their relevance to core issues in systems neuroscience, I think that the work will be of broad interest. (a) cells may code

for more than one feature, (b) development of models that capture this kind of behavior, (c) characterization of cells' behavior as a switching of modes, (d) demonstration of the functional importance of receptive field properties via a decoding analysis, and (e) pharmacologic analysis. The main concern is to what extent these components are conceptual advances. From my point of view, (d) and (e) clearly falls into this category, but (a) and (b) are well-established, and (c) requires further support. However, this is not to indicate that (a) and (b) are superfluous or detract from the paper – they need to be presented as a foundation for (c-e), but in a way that makes the distinction between established background and new material clearer.

With regard to (a), the authors have indeed cited many who take the view that retinal ganglion cells coded a single feature, or that their computations could be well-modeled by a simple spatiotemporally separable filter. But they also cite a number of previous authors that already showed that ganglion cell coding is more complex -- in some cases reaching back more than 30 years. One could reasonably add the work of Hochstein and Shapley (1976, see below), and Rodieck and Stone (J. Neurophysiol. 28, 832-849, 1965), who showed that center and surround have different dynamics – and thus, process inputs of different spatial scales with different timecourses. So I think it's fair to say that the idea that retinal ganglion cells only code one feature – although some in the field have adhered to it – is long past its expiration date.

With regard to (b), the model of nonlinear subunits is very similar to that advanced by Hochstein and Shapley for cat Y-type retinal ganglion cells (1976) (J. Physiol. 262, 265-284). That model has a linear center and surround, and an array of nonlinear subunits (H&S Figure 10). Notably, the array of nonlinear subunits extends to about four times the diameter of the center, so these cells would be expected to have the same behavior as the cells described in the present paper: linear responses that code for position in the center of the receptive field, and nonlinear ON-OFF responses sensitive to movement in the periphery. Later work (Victor and Shapley, 1979) added dynamics to this model, yielding a structure very similar to that of the authors' Figure 2D.

With regard to (c), the notion of “mode-switching” is interesting and novel. These earlier models did not contain such a component or functional interpretation; rather, the linear receptive field elements and nonlinear subunits combined by summation, followed by an output threshold simply due to the impossibility of a negative spike rate. But it is unclear whether the behavior shown in this paper requires something beyond this; further details below.

With regard to (d), the decoding analysis is very enlightening about the functional roles of the receptive field mechanisms. The results are clearcut. While one might argue that they are not surprising (based on intuitive considerations in the earlier papers), this does not diminish the importance of the clear and elegant demonstration of Figure 5. However, I think there is an interesting and important conceptual matter: while the results are clearcut at the level of the single neuron, it is unclear how they extend to an entire population. Put another way, if one decoded the population of OFF-cells that had their centers near the bar, would one also be able to recover speed? Conversely, if one decoded the entire population of OFF-cells that had their centers far from the bar, would one be able to recover position? This is especially relevant to the discussion on lines 266-273. Since the authors are using an array-recording technology, this would appear to be an accessible question.

With regard to the pharmacologic dissection (e), here also I think that the results are clearcut insofar as they demonstrate the pharmacologic properties of the mechanisms underlying the near- and far-bar responses. But I think they fall short of demonstrating active suppression, as doing so would require repeating the full modeling analysis on pharmacologically-manipulated preparations.

Related Specifics

Page 7, near top: do the recordings – either with the noise stimuli used to collect the definitive data, or even with simple hand-mapping -- provide a qualitative suggestion that the reason for the failure of the model is the presence of ON-OFF type responses? It would be helpful if there was some simple way of showing what the problem with the model is, to further motivate the next modeling step, and also to compare with established retinal physiology (see above references) concerning nonlinear subunits.

Page 8, bottom half: There may be a simpler explanation than active suppression that needs to be excluded: a simple threshold. The neurons shown do not appear to have a maintained firing rate, so it is reasonable to hypothesize that they only fire when some internal generator signal rises above threshold. If this is the case, the failure of additive combination of the responses to the near and far bars is not very surprising: each bar, by itself, produces transients that protrude above a baseline firing rate of 0 (Figure 2), but often (since the responses to the central bar is quasilinear), a bar will produce a negative internal “generator” signal. At these times, even though it is not producing any spikes, additive combination with the peripheral bar’s generator signal will appear to suppress it (as in Figure 3A). The fact that the two bars together produce a response that is very like the center alone (when measured as a correlation coefficient) is also expected based on the observation that the surround response is much smaller. So in sum, I think that in order to conclude that active suppression is happening, one must exclude the simpler model of linear summation followed by a threshold. Excluding the simpler model is a logical first step, and more convincing than showing that a suppressive model *can* fit the data.

In this regard, it is worrisome that the models provide a fairly imperfect fit to even the single-bar response (Figure 2B). Perhaps this is because the receptive field is a stylized Gaussian and the nonlinearity is a stereotyped nonlinear function ($\log(1+\exp(x))$), rather than sensitivity functions and nonlinearities that more accurately capture the responses (e.g., Nirenberg et al., 2010, 2012; Nichols, Nirenberg, and Victor 2013). Here the concern is that the authors are inferring that active suppression is needed because a suppression index calculated from the model is below the suppression index calculated from the neural data – but if the model does not even fit the neural data in the one-bar condition, this comparison may be misleading.

Page 14, lines 177-179: I think it is overstating the case to imply that there is a single visual feature that is extracted. For example, contrast and orientation matter too, both at the level of the single cell (contrast) and the population (both). So it’s a matter of characterizing the differences in what can be extracted, rather than identifying the single feature.

Page 19, ca line 257: The intended emphasis (w.r.t. “simultaneously”) is unclear. The context of the previous material would suggest that the authors mean that a cell can switch from coding one feature to coding another [as mentioned above, I am concerned that mode-switching is an overstatement], but here the term seems to mean that the same response train carries position and velocity information at the same time.

Minor

Page 5, 4 lines from bottom (and possibly elsewhere): quoting astronomically low p-values seems potentially misleading; it is highly unlikely that the assumptions underlying the statistical tests are so tightly obeyed so that $p < 10^{-25}$ is a meaningful value. Better to just say $p < 10^{-5}$ (or something), and quote the value of the t-statistic.

Page 22, line 320: AMES->Ames

Page 23, line 329-334: Maybe add an example of these trajectories to one of the figure panels? (Sorry if I missed it)

Page 26 (lines 399-401): it is unclear whether signals pass through two separate nonlinearities (the one on line 400 and then the one in eq. 4 (line 401), or whether eq. 4 is just a restatement of the text.

Page 28, line 437: Apparently some predicted PSTH's were negatively correlated with the model fit – the extent of these negative correlations should be mentioned here and in the figure legend for Figure 2F, as the authors indicate that it is suppressed in the graphical display. Presumably this only occurred for very peripheral bars, and the few points that are currently plotted on the axis.

Page 32: Please clarify that information was calculated based on actual responses, not a model, as there would be a major caveat if the model response was used.

Reply to reviewers

Reviewer #1 (Remarks to the Author):

Deny et al. combined experimental and computational modeling approaches to show that one type of retinal ganglion cells responds differently - and thus performs fundamentally different computations - when a light stimulus is presented inside or outside the receptive field center. The manuscript is written logically and the overall claim is supported by the results. As detailed below, however, I think that some critical data are missing to fully support it. The authors will then benefit a lot from further analyses and clarification.

=== Major points ===

(1) Linear versus nonlinear computations:

LN model is not "linear". The parameters enter linearly in the model (through the "static-nonlinear" link function), but the predictors (output responses) are nonlinear in relation to the input stimulus. It needs to be clarified what "computation" the authors are referring to. Or, it might be fair to say that the computation is "linearizable", rather than linear.

We agree with the reviewer that the LN model is not linear. Here we wanted to contrast it with the cascade model, which is much "more non-linear". We thank the reviewer for suggesting the term linearizable, which seems more relevant to describe the computation performed by the LN model. Instead of using the term "linear computation" we have now used "pseudo-linear" (a term employed by reviewer 3) or have directly mentioned the LN model.

line 97: show the pooling weights for the LN model in the figure. In general, please report all the parameter values in each model for each condition.

We have now shown the average pooling weights of the LN model as part of the supplementary figure 2. The pooling weights for the other condition are shown in figure 3.

line 98: the subunit models have quadratic rectifications. Summing up the outputs of ON and OFF subunits was then not linear, even though they have opposite response polarity and thus can contribute to "linearize" the inputs to a ganglion cell by cross-over inhibition (ie, "push-pull" circuits).

We agree that, with a quadratic rectification, the subunit model can never be strictly

equivalent to a LN model. For this, what would be needed is a “rectified linear” non-linearity with a threshold at 0.

For completeness we also tested a subunit model with linear rectification on our data. In the specific case of the central cells, it worked as well as a model with quadratic rectification ($r= 0.85 \pm 0.01$ for linear rectification vs $r=0.83 \pm 0.01$ for quadratic). However, the model with linear rectification for the subunits did not work as well as the quadratic one to predict the responses of distant cells ($r=0.53 \pm 0.04$ for linear rectification vs $r=0.73 \pm 0.02$ for quadratic rectification).

We think this idea of linearization of a subunit model has already been addressed in other papers (by Werblin and colleagues), and it is a minor point of our study. We decided to correct the text mentioned by mentioning that the push-pull organization of the subunit weights should help linearizing the effect of the stimulus, as suggested by the reviewer.

We have reformulated our explanation in the text.

(2) ON versus OFF subunit pathways:

The authors should analyze not just the ON subunit circuits but also the OFF subunit circuits. Off subunits have an order of magnitude stronger effects than On subunits (Fig 3), and thus I don't think it is reasonable to focus only on the On pathway. Moreover, Off subunit weights for the distant stimulus change the sign under the two conditions (supp fig 1): negative values in combined bar conditions, but positive values in single bar condition. (Note, in contrast, On subunit weights have antagonistic center-surround structure in both conditions.) This change in the Off subunit weights seems to have a stronger effect than that in the On subunit weights (I cannot judge from the hue plots in fig 3 and supp fig 1), and it seems to me that this is the key difference between the center and surround computations.

We agree with the reviewer that the emphasis on ON subunits was not properly justified, and that some analyses were missing to fully understand the respective contribution of On and Off subunits.

We have plotted the respective contribution of OFF and ON subunits against each other (see also reply to the next point) for the activation of distant cells, which shows that the distant responses are dominated by ON subunits for distant stimuli.

To further determine if distant responses are dominated by ON or OFF subunits, we plotted the responses to distant flashed bars predicted by the model. We found that the responses predicted by the model were largely dominated by ON responses.

The OFF subunits have therefore a strong contribution to central responses, but the responses to distant stimuli are mostly mediated by ON subunits.

We have improved the manuscript and added figures to better document the contribution of OFF subunits. We have also shown that the distant responses are dominated by the ON subunits in two different ways. These additional pieces of evidence have been added to the manuscript.

Figure 3: show the weights for the OFF subunits here (but not as a supplemental). Also, show the weights (or summary statistic) for ON and OFF subunits in the same scale

under the two conditions. This may be done better with line plots or bar graphs, but not with a hue.

We have now plotted a comparison with a line plot of OFF and ON subunits, showing that the positive weights of the OFF subunits are smaller than the ON ones in the surround.

Also, the gain control model should use the output of the subunit models, but not the data PSTHs, to examine how the gain changes for individual subunits. One can then analyze the gain control on the ON and OFF pathways separately. Similarly, data from pharmacology experiments should also be analyzed by using the subunit models to elaborate the ON and OFF circuits separately.

We have analyzed pharmacology experiments with our subunit model to better separate the ON and OFF circuits. We have performed additional experiments to fit our subunit model to data where a randomly bar is displayed when strychnine is applied (see below). We have fitted the subunit model to the responses of central cells. We observed that the weights of the ON subunits decreased to zero in this condition. In the normal condition, these weights were mostly negative, which is expected in the OFF ganglion cells we study here. Our interpretation is that these ON subunit weights correspond to a circuit where ON bipolar cells activate glycinergic amacrine cells, that inhibit the recorded ganglion cells, directly or through OFF bipolar cells. As a consequence, the blockade of these amacrine cells decreased the ON subunit weights to zero. We also observed that the negative weights of the OFF subunit filters, that were located in the near surround, disappeared when strychnine was applied. This suggests that these negative weights are mediated by a glycinergic amacrine cell.

We have added these results to the manuscript.

Fitting the gain control model is challenging because this is a non-convex optimization. For randomly moving bar stimuli, an additional challenge is the sparsity of responses to the distant stimulus, while responses to the central stimulus are much denser: when trying to fit the model to these responses, the optimization will give more weight to the denser responses. As a result, we did not manage to fit the gain control model to randomly moving bars, and this is why we used oscillating bars, where responses to central and distant stimulations can be separated. However, the oscillating bar stimulus does not allow fitting the subunit model. We would therefore need to record the same cells with both the randomly moving bar stimuli to learn the subunit model, and the oscillating bar stimulus to learn the gain control model. Unfortunately, it is impossible to obtain a stable recording that can allow displaying the two stimuli. We therefore used the PSTHs as a proxy for the gain control model. The main difference between using the weighted sum of subunit outputs and the PSTH is the last non-linearity of the model. We have added data to show that this non-linearity cannot explain the suppression observed.

We have explained this limitation in the manuscript, and have added data and analyses to show that the last non-linearity of the model cannot explain the suppression observed.

(3) Disinhibitory circuit:

Experimental data are too weak to fully support disinhibitory circuits. All the authors showed was that antagonistic surround response to a flashing bar (why not using the randomly moving bars?) is suppressed by strychnine. Disinhibitory circuits may work for ON subunit pathway, but not for OFF. How is each of the ON and OFF pathways affected?

As we showed above, the response in the surround are mostly dominated by ON inputs, and OFF responses are either weak or inexistent. Nevertheless, to estimate how distant responses are affected by strychnine, whether they are ON or OFF, we conducted additional experiments where we displayed a randomly moving bar with and without strychnine. All the responses of distant cells were suppressed when strychnine was present. This shows that both ON and OFF pathways are suppressed by the strychnine, and suggests that both ON and OFF distant responses are mediated by a glycinergic pathway. We have also conducted other experiments to better understand this disinhibitory pathway, that are detailed in the reply to the next questions of the reviewer.

We have considerably enriched the figure where the effect of strychnine was described to show with additional panels that it blocked the responses to a randomly moving bar, and have added some text to describe this with more details. .

Also, what is the difference between the "classic" antagonistic surround responses and the distant responses the authors are reporting here? Classic explanation of antagonistic surround responses is through lateral inhibition via horizontal cells. Is there any contribution of such circuits? Also, is there any contribution of gap junctions? The pharmacology experiments with randomly moving bars, instead of flashing bars, will be more suitable to compare the changes of the parameter values in the subunit model.

The horizontal pathway does not involve any glycinergic receptor. This suggests that horizontal cells do not mediate these distant responses. To test if gap junctions are involved in generating these distant responses, we used the blocker 18-bGa. This blocker is known to block specifically gap junctions but can have off-target effects after a prolonged exposure, that affect the visual responses (Trenholm et al, Nature Neuroscience, 2013). To allow recording many instances of the same stimulus, we decided to record the responses to distant flashes. We found that the gap junction blocker decreased the distant responses in some cases (figure 1A, B), but the effect was not significant over the entire population – the ratio of cells responding to the bar even increased (figure 1C). Central responses were preserved. This suggests that gap junctions may help propagating the activity laterally, but are not necessary for distant responses.

Figure 1: A: Response of one cell to a bar flashed inside the receptive field center. Black raster: control. Blue raster: after the application of gap junction blocker 18 β -glycyrrhetic acid (25 microM). B : same as A, but for a bar outside of the receptive field center. C : percentage of cells responding to a central flash (left) and to a distant flash (right) before and after 18 β -glycyrrhetic acid application.

Our experiments with strychnine demonstrate that glycinergic amacrine cells are a necessary component to generate distant responses. It is known that some glycinergic amacrine cells are reciprocally connected by gap junctions. We have previously drawn the hypothesis that the distant responses are due to a disinhibitory loop of glycinergic and GABAergic amacrine cells. Our hypothesis to explain this partial sensitivity to gap junction blockade is that glycinergic amacrine cells are activated by the distant visual stimulation, and this activation is further amplified by activating neighbor glycinergic amacrine cells through gap junctions. When blocking gap junction, this amplification is lost. The distant responses are therefore weaker but still visible.

As detailed in the reply to the previous point, we have now conducted additional experiments where we measured the response of ganglion cells to a randomly moving bar when strychnine was added to the bath. Since all the responses of distant cells were completely abolished, the corresponding parameters in the subunit filters are zero.

We have mentioned the results about blocking gap junctions, and added the results of fitting the model to responses under strychnine.

Glycinergic amacrine cells have a narrow field. Wide-field GABAergic amacrine cells are most likely required if indeed disinhibitory circuits underlie the reported phenomena. But, if push-pull mechanisms are involved in the center response (to "linearize" the

synaptic inputs to ganglion cells), why there is no change in either the evoked or baseline firing after applying strychnine (fig 6A,C)? Narrow-field glycinergic amacrine cells are most likely involved here, and I expect that strychnine affect these cells.

We agree with the reviewer that glycinergic amacrine cells are narrow field and cannot explain alone the distant responses. We hypothesized that wide field GABAergic amacrine cells could also be involved in the process. We therefore tried to block GABA receptors to test if this affects the distant responses. Note that when using picrotoxin (or other GABAergic blockers), it is very hard to obtain a stable recording window. After some time, it is well documented (Demb et al, 1999) that large oscillations appear that prevent the study of any visual response. This is also what we found in our recordings (figure 2).

Figure 2: Oscillations due to picrotoxin. Activity of one cell over time. Each line corresponds to a window of activity of 5.5 s, time goes from bottom to top. In this experiment, oscillations appeared 10 minutes after adding 10 microM of picrotoxin to the bath.

To avoid these large oscillations that considerably affect the visual responses, we used a small concentration of picrotoxin. This concentration allows us to record over a time window where picrotoxin affects the activity, before the onset of the large oscillations. During this period, we flashed bars at various distances.

We observed that distant responses were selectively decreased, while central responses remained unchanged. Spontaneous activity also increased.

The short duration of the window where the drug was effective at blocking inhibition, but before the onset of oscillations, only allowed us recording responses to distant flashes, and did not give us enough time repeat the randomly moving bar experiment. However, since our results on blocking the glycinergic receptors showed similar effects for flashed bars and randomly moving bars, we think it is likely that the same circuit is involved in both cases.

Our results strongly suggest that distant responses are mediated by a disinhibitory loop of GABAergic and glycinergic amacrine cells.

We have added these results in the paper.

Finally, while we did not observe a significant change in the spontaneous activity when blocking the glycinergic receptors with strychnine, we observed changes in the push-pull mechanisms of the central responses. We have fitted our subunit model to responses of central cells to a randomly moving bar, after application of strychnine. We observed that the weights of the ON subunits decreased to zero in this condition. In the normal condition, these weights were mostly negative, which is expected in the OFF ganglion cells we study here. Our interpretation is that these ON subunit weights correspond to a circuit where ON bipolar cells activate glycinergic amacrine cells, that inhibit the recorded ganglion cells, directly or through OFF bipolar cells. As a consequence, the blockade of these amacrine cells decreased the ON subunit weights to zero.

We have added the results of these novel experiments and analyses in the paper.

(4) Methods clarification:

Details are missing from the methods section. It is critical to describe everything in the methods; otherwise no one else can reproduce the results.

Typing: Need more details. How many types did the authors obtain? How did the authors classify kinetics (FAST vs SLOW) within OFF type cells? Also, among the previously reported cell types, which is it that (most likely) correspond to the one reported here?

The response to a flash was transient, and there was only a response to a full field flash when light decreased, not when light increased. The type we report here corresponds most likely to an OFF alpha transient type. The average receptive field size was 270 microns. When comparing to previous classifications performed on mouse ganglion cells based on anatomy, the type we studied mostly likely correspond to the G3, G7, G11 or G18 types described by Volgyi et al (2006). Compared to the types described in the mouse by Baden et al (2016), this type is closest to the OFF alpha transient (8a or 8b in their classification).

We have added this clarification in the paper.

Subunit models: are there 100 subunits each for ON and OFF subunits (line 385)? The Eq.(3) seems to define only OFF subunits, but what is the definition of the ON subunits? In particular, do the authors use different parameters between ON and OFF subunits, for example, to take into account that ON responses are generally slower than OFF responses? Also report the obtained parameters values (alpha, beta, theta) after fitting.

Yes, there are 100 OFF subunits and 100 ON subunits. The ON subunits are identical to OFF subunits, except that the filter was multiplied by -1. We did not take into account that ON responses are generally slower than OFF ones at the level of the subunit, but the second filter, which is fitted on the data, integrates the output of the subunits with several time lags and can take this into account.

We have clarified this in the methods and also reported the values obtained for alpha, beta and theta.

Fitting: is fitting done for each cell? I assume that the reported weights are obtained from population data (because moving bars jitter only a limited range), but would it be beneficial to accumulate population data first and perform data fitting (rather than doing data fitting first and accumulate the parameters to analyze the weights as in figure 3)?

The fitting is done for each cell. Doing the fit on the entire population at once would be technically challenging as it would require estimating the likelihood over the entire population activity, and this would have a high computational cost. Also, it is not clear what would be the advantage of doing this instead of fitting each cell separately.

We have clarified that fitting is done for each cell in the methods.

Gain control model: the authors should use the subunit model outputs for r_c and r_s in Eq.(11), instead of the data PSTHs, especially for the r_s (see above). In particular, if the authors claim that the subunit model should be used to explain the distant stimulus response, then would it make more sense to examine the gain change for the output of those subunits separately? Also, report the fitted parameter values (α_0 , α_1 , β_0 , β_1 , and H). Also, any idea how the gain control is implemented?

As we explained above, there are technical limitations in data acquisition and fitting that prevented us to use subunit outputs instead of PSTHs. As a result, we could not fit a gain control with different gain changes for the different subunits. The main difference is that the subunit outputs, after being filtered, are passed through a non-linearity to predict the PSTH. We have shown that this second non-linearity cannot explain by itself the suppression of the bar.

We have explained this in the text. We have reported the values of the parameters.

The gain control can be implemented in several ways, which are not mutually exclusive. One possibility is that gain control is shaped by the intrinsic properties of ganglion cells, and in particular the slow inactivation of sodium channels (Kim and Rieke, 2003). Another possibility is a feed-forward inhibition from amacrine cells (Zahgloul et al, 2007). Both mechanisms have been proven to control the gain in ganglion cells.

We have added these possible implementations of gain control in the text.

=== Minor points ===

(1) Page 1, last line: better to say stimulus "location".

We have corrected this.

(2) line 30: Say "Individual cells switched from this computation", instead of "one

computation"

We have corrected this.

(3) lines 79-80. This is going too far. The LN works well for a randomly moving stimulus, but it was not shown that it works well for a general stimulus, say, white-noise, natural stimulus, etc.

We agree that we have only shown that the model works well for a bar (and for a moving texture, a novel result we have added to answer reviewer 2's question).

We have corrected this sentence.

(4) Figure 1: How about the distant cells on the other side (Fig 1E)? Are cells on the opposite sides also responding synchronously?

In the synchrony analysis we took cells that were sufficiently far from the bar, from each side of the bar. However, in most cases we had only a couple of neurons from the upper side.

(5) Figure 1D: Show all data points from individual pairs. How about central-distance cell pairs? Also, how about correlation at the spike level (finer time resolution) instead of PSTH?

We have shown all data points in the figure 1D.

Correlation at the spike level, with the same resolution than for the PSTHs (time bin=17ms), gave a quite similar result than with the PSTHs (distant cells are synchronous across longer distances than central cells, figure 3), albeit less striking than with the PSTHs. This difference can easily be explained by the fact that the underlying firing rates (that we estimate when we compute PSTHs) are more reliable than individual spike trains. Central-distant pairs yielded an intermediate curve between the two other curves (in green below). We observed that central cells were not as correlated with distant cells at long distances as distant cells between themselves.

Figure 3: correlation at the spiking level for pairs of cells, as a function of their pairwise distance measured along the bar motion axis, shown separately for cells whose receptive field center either was (blue) or was not (red) stimulated by the bar. Curves: average values +/- SE. Green: central-distant pairs.

(6) Line 170. "The subunit model equipped with gain control" This is not directly tested. The methods section says that the gain control model uses data PSTHs, but not the ones from subunit models.

We agree with this remark and have modified the text to say that such a model has the potential to explain all the responses recorded here.

(7) Line 171. Response to "randomly moving bar" was not tested. According to the methods section, it was sinusoidal waves that the authors used to test the gain control model.

We agree with the reviewer and we have toned down this sentence.

(8) Figure 4: how does the random gain control model work for randomly moving bar stimuli?

As we explained above, we could not fit the model directly to the randomly moving bar stimuli. Thanks to our work on fitting the gain control model to oscillating bar data we could show that a gain control was present in these cells and could explain the suppression observed. We have modified the text to clarify this.

(9) line 198: Change "Fig 3" to "Fig 3C"

We have corrected this.

(10) line 203: remove comma

We have corrected this.

(11) line 337: what is the bar length?

The bar length is 2mm. We have corrected this.

(12) line 417: spell out BFGS (Broyden-Fletcher-Goldfarb-Shanno)

We have corrected this.

(13) line 479: what is the bar length? Also, what is the amplitude of the oscillating sign wave?

The bar length is 2mm and the amplitude of the oscillating sinewave 350 microns. We have added this in the methods.

(14) Eq.(10): integral should be from $t-\tau$ to t (remove "prime" on the top t).

We have corrected this.

(15) line 497: show data to support the claim (quadratic dependence of the PSTH).

We have added data showing that there was a (weak but existent) quadratic dependence of the PSTH in fig 4C.

(16) Eq.(2) in supplement: use three dots to represent the range: $\tau=T...T+DT+L$

We have corrected this.

(17) line 31 in supplement: should read "same property as the one found on"

We have corrected this.

Reviewer #2 (Remarks to the Author):

In their study, Deny et al. experimentally characterize and model the receptive field organization of a transient OFF-RGC type (OFFT-RGCs) in rat retina, and trace the influence of this receptive field organization on population coding of motion information. In particular, the authors show that receptive field center and surround of these cells are driven by nonlinear subunits, predominantly OFF-signed in the center and ON-signed in the surround. Due in part to nonlinear spatial integration, OFFT-RGCs respond to motion in both the center and the surround. Interestingly, the authors show that responses to surround motion are suppressed by center stimulation, which they explain by a global gain control mechanism implemented in a cascade model after spatial summation. Deny et al. further show that responses to motion in the receptive field center encode the position of a moving object, whereas responses to distant motion encode the speed of a moving object. This implies that at any particular time different OFFT-RGCs in a population encode distinct features of motion. Finally the authors provide pharmacological evidence that responses to movement of distant stimuli are mediated by a disinhibitory circuit involving glycinergic amacrine cells.

Overall this is an interesting study. The experiments appear carefully conducted, and their motivation and results are explained clearly in the text and illustrated well by the figures. It is in many ways quite a traditional multi-electrode exploration of visual processing in the retina, and shares some of the weaknesses of this approach. By the standards of modern neuroscience, the understanding of circuit mechanisms underlying particular responses remains vague, as recordings are limited to spikes of the output neuron of the circuit. The fact that responses can be described by a model is orthogonal to experiments that would e.g. record from presynaptic neurons to test specific mechanisms of computation. Moreover, while one can show that different parts of a neuronal population can encode different parameters of motion (position vs. speed), there is no evidence that / how the brain uses this information. These criticisms are not unique to the present study, and the authors do not overstate or -interpret their findings. I mention them nonetheless, because given the inherent limitations of the

approach, it is particularly important that the authors provide clarify the robustness, specificity, and context of their findings to warrant publication in a top-tier journal.

We thank the reviewer for his comments and agree that multi-electrode arrays give only limited insight on the circuit mechanisms. However, we think it is still a valuable technique, which gives complementary insights to single cell intracellular recordings. The key advantage of large-scale multi-electrode array recordings is the ability to record from hundreds of neurons simultaneously. Here we extracted from these recordings the activity of ganglion cells of the same type, and could therefore record the activity of many cells of the same type *simultaneously*. As a result, we could see how this homogeneous population can compute two different features on the same stimulus. Because we have access to the activity of neurons at the same time, we could see that, although the neurons distant from the bar had weaker responses to the stimulus, these responses were highly synchronous. As a result, a downstream neuron that would integrate over this population would be activated by these synchronous inputs. We would not have been able to demonstrate this directly with single cell recordings. We also agree also that we can only speculate about how the retinal outputs will be integrated by the brain. Recent works have shown that single cell in the LGN can integrate over many ganglion cells (Rompani et al, 2017). It might be possible for some LGN cells to detect the synchronous activity of many ganglion cells.

We have added these arguments in the discussion.

Another argument in favor of this technique is that the kind of modeling we did here, which allowed understanding what computation is performed by these cells, require very long recordings, typically a couple of hours of stable activity. This is rarely achieved with intracellular recordings.

Multi-electrode array recording is therefore a technique complementary to intracellular recordings, that allow to address different but important questions. As acknowledged by the reviewer, many other studies have similar limitations.

We also agree that multi-electrode array gives a limited insight on the mechanisms of these distant responses. We have added more data from novel pharmacology experiments in this revision that further support our hypothesis of a disinhibitory circuit. To further elucidate the mechanisms of this distant activation, we would have to target specific subtypes of amacrine cells genetically and perform intracellular recordings of both these amacrine cells and ganglion cells simultaneously to demonstrate their implication. These challenging experiments are clearly beyond the scope of this study, but we mention this possibility in the discussion.

We have mentioned this in the discussion.

Specific comments

1) It is important to clarify to what extent responses to center and distant motion, and the suppressive effects of center motion on distant stimuli are specific to OFFT-RGCs or

are observed in other ON, OFF and ON-OFF RGCs. The authors should include data from other cell types to illustrate differences or similarities in their motion responses and receptive field architectures.

We agree with the reviewer that describing other types of cells could be interesting and beneficial to the paper. We have performed recordings from another type of cells, where we could clearly isolate another single type in our data. This other type was most likely ON alpha cells. In this type, we observed responses to distant bars, and suppressive effect of the center, but with a striking difference compared to the OFF type: both central and distant responses were non-linear. We could not predict the responses to a bar moving inside the receptive field with a LN model, opposite to the OFF type where stimuli were processed linearly in the receptive field center. Our subunit model was able to predict these central responses, as well as distant responses.

In this type, the similarity was that cells responded to distant stimuli, that these distant responses were suppressed when the receptive field center was stimulated, but the receptive field architectures were very different. This emphasizes that the specificity of the OFF type is to mix two different computations: a central computation that is linear, and a distant computation that is non-linear. In contrast, this ON type seems to perform a similar computation in the center and the surround.

We have added a figure and a corresponding text in the results section to describe these ON cells.

2) How robust is the described receptive field architecture and motion encoding to stimulus conditions. The authors should include data from scotopic stimulation in addition to their current photopic data. This would help clarify whether motion information available to the brain via OFFT-RGCs is similar or qualitatively different depending on ambient illumination. There is some indication in the literature that transient OFF RGCs in rodents switch responses as a function of luminance (Farrow et al. Neuron 2013).

We have conducted additional experiments to test if the effects we described were present for different luminance levels. We varied the luminance from 10^9 to 10^{13} photons/cm/s, across 5 different levels. In all cases, we found that there were distant responses. A LN model could explain the responses to stimulation of the receptive field center, but not distant responses, confirming our results over a broad range of light levels. We also observed that the linear filter in the LN model estimated changed at each light level, confirming previous reports, like Farrow et al, 2013. To record the same cells at different light levels we had to reduce the duration of the recording for each light level. For this reason, because distant responses were weaker at lower light levels, and because spontaneous activity was larger, we did not have enough data to fit our subunit model at each light level. These results still allow us to claim that there is a pseudo-linear computation when the stimulation is inside the receptive field center, and a non-linear computation for distant stimuli. The model architecture is therefore valid at least qualitatively over a broad range of light levels.

We have added a supplementary figure describing these results.

3) The receptive field model that the authors present accurately predicts responses to stimuli of the sort used in its construction. It is important for the authors to show whether their model generalizes e.g. to predict motion of textures or patterns rather than bars.

We have performed additional experiments where moving textures were displayed in different parts of the visual field. We found that a similar model to the one we used for the bar predicted well the responses to textures. The only difference in the model that we had to use was that the non-linearity of the subunits had to be changed compared to what we used for the bar stimulus. Nevertheless, all the salient features of the model architecture were maintained and it performed well in predicting the responses to this randomly moving texture.

We have added these results in the paper, which show that our model can also work for randomly moving textures.

4) Regarding context. Transient OFF RGCs have frequently been characterized in mice and have been identified as a particular molecular and morphological type in mice (Huberman et al. Neuron 2008, Munch et al. Nat Neurosci 2009, Farrow et al. 2013, Murphy et al Neuron 2006...). The authors should attempt to establish whether the cells they record in rats are a homologous type by comparing characteristic response parameters.

The response to a flash was transient, and there was only a response to full field flash when light decreased, not when light increased. The type we report here corresponds most likely to an OFF alpha transient type. The average receptive field size was 270 microns. When comparing to previous classifications performed on mouse ganglion cells based on anatomy, the type we studied mostly likely correspond to the G3, G7, G11 or G18 types described by Volgyi et al (2006). Compared to the types described in the mouse by Baden et al (2016), this type is closest to the OFF alpha transient (8a or 8b in their classification).

We have added this clarification in the paper.

Reviewer #3 (Remarks to the Author):

Multiplexed computations in retinal ganglion cells of a single type

The authors record and analyze large-scale multineuronal spiking activity in the ganglion cell layer of the rat retina. They interpret their findings as showing that a single

cell type (fast OFF ganglion cells) codes in two different ways: (i) a quasilinear mode that dominates for objects that are near the receptive field center and represents position, and (ii) a nonlinear mode that dominates for objects that are far from the receptive field center and primarily represents speed.

There are several findings and contributions reported in this paper, and, because of their relevance to core issues in systems neuroscience, I think that the work will be of broad interest. (a) cells may code for more than one feature, (b) development of models that capture this kind of behavior, (c) characterization of cells' behavior as a switching of modes, (d) demonstration of the functional importance of receptive field properties via a decoding analysis, and (e) pharmacologic analysis. The main concern is to what extent these components are conceptual advances. From my point of view, (d) and (e) clearly falls into this category, but (a) and (b) are well-established, and (c) requires further support. However, this is not to indicate that (a) and (b) are superfluous or detract from the paper – they need to be presented as a foundation for (c-e), but in a way that makes the distinction between established background and new material clearer.

With regard to (a), the authors have indeed cited many who take the view that retinal ganglion cells coded a single feature, or that their computations could be well-modeled by a simple spatiotemporally separable filter. But they also cite a number of previous authors that already showed that ganglion cell coding is more complex -- in some cases reaching back more than 30 years. One could reasonably add the work of Hochstein and Shapley (1976, see below), and Rodieck and Stone (J. Neurophysiol. 28, 832-849, 1965), who showed that center and surround have different dynamics – and thus, process inputs of different spatial scales with different timecourses. So I think it's fair to say that the idea that retinal ganglion cells only code one feature – although some in the field have adhered to it – is long past its expiration date.

We agree with the reviewer that previous works have already cast serious doubts on the “single feature” idea. However, we think this idea has been revived with the recent works where some types of ganglion cells could be very well isolated using either genetic markers (Da Silveira and Roska, 2011), or systematic clustering of physiological responses (Baden et al, 2016).

The influence of the surround has been described by many studies. Many works have described how the surround can modulate the response to the center, and some works have focused on how it can generate responses by itself. The question is whether the responses that are generated purely by the surround can be considered as qualitatively different from the one generated by the center, or if this is the same computation carried out in both the center and the surround, albeit with a longer delay and/or a smaller influence on the firing rate. The answer to this specific question is not always entirely clear in these studies.

We have added additional data from another type of ganglion cell, showing that the two are possible. In the type we focused on, we can see a linear computation in the center, and a non-linear one in the surround, explainable only by the subunit model. However, we found another type of ganglion cells where the same non-linear computation, predicted by the subunit model, was present both in the center and in the surround.

Furthermore, the influence of the surround could be interpreted in two different ways. One possibility is that the surround sensitivity is here to maintain the same selectivity to a given feature in different visual contexts.

Our results suggest an alternative, where surround selectivity is used to transmit information about another feature through a different channel, i.e. with synchronous activity.

To conclude, we agree with the reviewer that the “one feature” idea, at least in its simplest form, has been seriously questioned, but we think there is a long standing debate here, and we think our study is an interesting contribution to this debate.

With regard to (b), the model of nonlinear subunits is very similar to that advanced by Hochstein and Shapley for cat Y-type retinal ganglion cells (1976) (*J. Physiol.* 262, 265-284). That model has a linear center and surround, and an array of nonlinear subunits (H&S Figure 10). Notably, the array of nonlinear subunits extends to about four times the diameter of the center, so these cells would be expected to have the same behavior as the cells described in the present paper: linear responses that code for position in the center of the receptive field, and nonlinear ON-OFF responses sensitive to movement in the periphery. Later work (Victor and Shapley, 1979) added dynamics to this model, yielding a structure very similar to that of the authors’ Figure 2D.

We agree with the reviewer that our model bears some similarity with the model of Hochstein and Shapley (1976), and of Victor and Shapley (1979). In both cases the model is composed of a first layer of subunit, which filter the stimulus and pass the result through a non-linearity. Then a second layer is again composed of a filter and a non-linear function.

The models are therefore similar. However, we think there are some interesting differences. In the case of Hochstein and Shapley, the subunits covered both the center and the surround (fig 3 and 10), and there was a non-linear response even to center stimulation that could only be explained by the subunit model. In contrast we found that the responses to central stimulations could be explained by a pseudo-linear model.

But other types of cells show a different behavior: we have found another type of ON cells where non-linear responses were largely dominant both in the center and the surround: for both central and distant stimulations, the LN model did not perform well, and the subunit model could predict the responses.

We therefore conclude that there are two possible configurations for the subunit model: either the responses to central stimulation are amenable to a LN model, like the OFF type we focused on, or they remain non-linear, like the ON type we added in the revision.

From the results of Hochstein and Shapley, it is clear that responses in the surround are non-linear, but the relative importance of linear and non-linear responses in the center is not completely clear. Based on our own results, our hypothesis is that in their study, they may have pooled together different cell types with different behaviours.

A consequence of this difference is the relative importance of ON and OFF subunit weights in our model. In the center, both ON and OFF subunits contribute, with opposite signs, and can be approximated by a linear filtering. In the surround, the ON subunits have a much larger contribution than the OFF. To our understanding, this change of balance between ON and OFF relative weights in the center and surround was not reported in Hochstein and Shapley.

We have included novel data to contrast the multiplexing of two different computations we found for this fast OFF type with another well isolated ON cell type where the same non-linear computation was performed both in the center and in the surround. We have also discussed more in depth the comparison with previous literature.

With regard to (c), the notion of “mode-switching” is interesting and novel. These earlier models did not contain such a component or functional interpretation; rather, the linear receptive field elements and nonlinear subunits combined by summation, followed by an output threshold simply due to the impossibility of a negative spike rate. But it is unclear whether the behavior shown in this paper requires something beyond this; further details below.

We have tested this simpler model and have added in the paper several lines of evidence (detailed below, in reply to specific comments) that advocate against this model.

With regard to (d), the decoding analysis is very enlightening about the functional roles of the receptive field mechanisms. The results are clearcut. While one might argue that they are not surprising (based on intuitive considerations in the earlier papers), this does not diminish the importance of the clear and elegant demonstration of Figure 5. However, I think there is an interesting and important conceptual matter: while the results are clearcut at the level of the single neuron, it is unclear how they extend to an entire population. Put another way, if one decoded the population of OFF-cells that had their centers near the bar, would one also be able to recover speed? Conversely, if one decoded the entire population of OFF-cells that had their centers far from the bar, would one be able to recover position? This is especially relevant to the discussion on lines 266-273. Since the authors are using an array-recording technology, this would appear to be an accessible question.

We thank the reviewer for asking this interesting question. It is possible that, even if a single distant cell gives little information about the position of the bar, it does not predict how well position can be decoded when reading out the activity from several of these distant cells. To address this issue, we have performed linear decoding of the position of the bar using different groups of cells. We found that, when using distant cells to decode the position of the bar, performance was always much lower than when using central cells. This difference in performance was maintained for up to 10 cells, the largest number of distant cells we could record simultaneously. Since the decoding performance increased with the number of cells, it is likely that, if we were to decode from an extremely large population of distant cells, we could recover the position of the bar, but this would require many more cells than when decoding from central cells.

We have added this new analysis in the results.

With regard to the pharmacologic dissection (e), here also I think that the results are clearcut insofar as they demonstrate the pharmacologic properties of the mechanisms underlying the near- and far-bar responses. But I think they fall short of demonstrating active suppression, as doing so would require repeating the full modeling analysis on pharmacologically-manipulated preparations.

We have done the full modeling analysis on pharmacologically-manipulated preparations. Adding strychnine in the bath to block the transmission of glycinergic amacrine cells blocked the distant responses. As a result, when fitting the subunit model to the data acquired under strychnine, there was simply no influence of distant subunits. Inside the receptive field center, adding strychnine suppressed the influence of ON subunits and left the influence of OFF subunits unchanged in the center, while removing negative weights in the near surround. Our interpretation is that these OFF ganglion cells receive direct input from OFF bipolar cells, and ON bipolar cells activate glycinergic amacrine cells that inhibit the ganglion cells, or bipolar cell inputs to these ganglion cells. The distant responses are also conveyed through glycinergic amacrine cells. We have performed additional experiments to show that this distant activation depends critically on GABAergic transmission, and not on gap junction. These experiments allow us clarifying the circuits that mediate the distant responses.

We have added these results in the paper.

Related Specifics

Page 7, near top: do the recordings – either with the noise stimuli used to collect the definitive data, or even with simple hand-mapping -- provide a qualitative suggestion that the reason for the failure of the model is the presence of ON-OFF type responses? It would be helpful if there was some simple way of showing what the problem with the model is, to further motivate the next modeling step, and also to compare with established retinal physiology (see above references) concerning nonlinear subunits.

From the responses of these OFF ganglion cells to full field flashes and flashed bars, we could determine that the responses to center stimulation was purely OFF: there were no ON responses. For surround stimulation, the ON component was largely dominating over the OFF one, and this was confirmed in the subunit model. We did not see any evidence that the failure of the LN model was due to the presence of ON-OFF responses.

We have added these results in the paper, and we have also expanded the discussion to compare with the references mentioned above.

Page 8, bottom half: There may be a simpler explanation than active suppression that needs to be excluded: a simple threshold. The neurons shown do not appear to have a maintained firing rate, so it is reasonable to hypothesize that they only fire when some

internal generator signal rises above threshold. If this is the case, the failure of additive combination of the responses to the near and far bars is not very surprising: each bar, by itself, produces transients that protrude above a baseline firing rate of 0 (Figure 2), but often (since the responses to the central bar is quasilinear), a bar will produce a negative internal “generator” signal. At these times, even though it is not producing any spikes, additive combination with the peripheral bar’s generator signal will appear to suppress it (as in Figure 3A). The fact that the two bars together produce a response that is very like the center alone (when measured as a correlation coefficient) is also expected based on the observation that the surround response is much smaller. So in sum, I think that in order to conclude that active suppression is happening, one must exclude the simpler model of linear summation followed by a threshold. Excluding the simpler model is a logical first step, and more convincing than showing that a suppressive model *can* fit the data.

We agree that we did not exclude this simpler model in the previous version of the manuscript. We tested if this model of a linear summation could predict the suppression observed in the data, with two different strategies, a “model-independent” method and another based on fitting models to the data.

In the first strategy we performed a test directly on data, without relying on models fitted on the data. Following the explanation described above by the reviewer, if the central stimulation evokes even a moderate level of spiking activity at a time where there is a response to the central stimulation, then we know that there is no negative generator signal at this time, and the distant response should not be suppressed.

We have collected responses of these OFF ganglion cells to randomly moving textures, and for this stimulus we found several clear examples where, despite a moderate level of spiking activity triggered by the central stimulation, the response to the distant stimulation was suppressed. We confirmed these observations by estimating our suppression index only in the time windows where there was a minimum level of spiking activity (note that this suppression index does not rely on the fitted model – see the reply to the next point).

In a second strategy, we used the models fitted on the central stimulation and the distant stimulation alone. The second filter applied to the subunit outputs gives a prediction of what this internal generator signal should be. We thus obtained two internal signals, one of the central stimulation, one for the distant stimulation. We then summed them and then passed the result through the non-linearity to predict the PSTH. We found that this gives a rather poor prediction of the PSTH and, more importantly, that it largely underestimated the suppression index measured on the data. We obtained this result for both the randomly moving bars and the randomly moving textures.

These results exclude the simpler model of a linear summation followed by a threshold, or at least make this explanation highly unlikely. We have reported them in the manuscript.

In this regard, it is worrisome that the models provide a fairly imperfect fit to even the single-bar response (Figure 2B). Perhaps this is because the receptive field is a stylized

Gaussian and the nonlinearity is a stereotyped nonlinear function ($\log(1+\exp(x))$), rather than sensitivity functions and nonlinearities that more accurately capture the responses (e.g., Nirenberg et al., 2010, 2012; Nichols, Nirenberg, and Victor 2013). Here the concern is that the authors are inferring that active suppression is needed because a suppression index calculated from the model is below the suppression index calculated from the neural data – but if the model does not even fit the neural data in the one-bar condition, this comparison may be misleading.

First, we want to clarify the definition of the suppression index: the suppression index does not rely on the fitted model, and is simply based on the correlation between the response to the distant bar and the response to the two bars minus the response to the central bar. This is a way to quantify if the response to the distant bar is still present in the response to the two bars.

The suppression for the linear model was a misnomer; here the purpose is to test if the suppression index could be explained by the noise in the response. We have corrected this in the method and the text.

The estimation of the suppression index does not depend on the model fitted to the single bar response. Nevertheless, we tested if changing the non-linearity could better capture the response to the central bar, using a similar approach to Nirenberg et al.

We parametrized the non-linearity as a cubic spline. We varied the different parameters of this non-linearity (number of knots in the cubic spline, bin size) to maximize performance. To fit this non-linearity to the data, we adopted the same strategy as Nirenberg et al (2010): starting from the filter inferred with the $\log(1+\exp())$ non-linearity, we alternated between fitting the cubic-spline non-linearity while holding the filter fixed, and inferring the filter for a fixed non-linearity. In both cases we did the inference by maximizing log-likelihood.

We tested this model on the responses to the stimulus with two bars. We found that this more flexible non-linearity added nothing significant to the prediction performance (figure 4A). We plot one representative example where the non-linearity obtained with the inference is compared with the $\log(1+\exp())$ function (figure 4B). There was little difference between them. We also plotted the output of the filter against the average firing for each time bin of a repeated stimulus sequence. The shape of the non-linearity follows these points, which suggest that the non-linearity was correctly fitted to the data.

These results suggest that $\log(1+\exp())$ was a satisfying model for the non-linearity, which explains why we did not see a significant improvement in performance with a cubic spline non-linearity.

Figure 4. A: performance in predicting the responses to the stimulus with two bars using a cubic spline non-linearity (y-axis) or a $\log(1+\exp())$ non-linearity (x-axis). B: example of a non-linearity obtained for one cell. Red: cubic spline non-linearity. Black: $\log(1+\exp())$ non-linearity. Blue points: average firing rate for one time bin against the filter output. Each point corresponds to a different time bin.

These results did not seem compelling enough to be presented in the paper, but we mentioned that we tried this cubic spline non-linearity and that it gave similar performance.

Page 14, lines 177-179: I think it is overstating the case to imply that there is a single visual feature that is extracted. For example, contrast and orientation matter too, both at the level of the single cell (contrast) and the population (both). So it's a matter of characterizing the differences in what can be extracted, rather than identifying the single feature.

We agree with the reviewer. We have changed the text to say that the cells are sensitive to the exact position of the stimulus.

Page 19, ca line 257: The intended emphasis (w.r.t. "simultaneously") is unclear. The context of the previous material would suggest that the authors mean that a cell can switch from coding one feature to coding another [as mentioned above, I am concerned that mode-switching is an overstatement], but here the term seems to mean that the same response train carries position and velocity information at the same time.

Here we aimed at saying that the entire population of cells of a single type carries position and velocity information. A single neuron does not carry both at the same time. Here our point was to emphasize that, within this single type of cells, a subset of cells carry more information about position, and another, disjoint subset carries much less information about position. Although this might seem a trivial point to the reviewer, we think this is worth emphasizing because many recent studies view cells of the same cell type as an ensemble that is functionally homogeneous.

We have modified the text to clarify this point.

Minor

Page 5, 4 lines from bottom (and possibly elsewhere): quoting astronomically low p-values seems potentially misleading; it is highly unlikely that the assumptions underlying the statistical tests are so tightly obeyed so that $p < 10^{-25}$ is a meaningful value. Better to just say $p < 10^{-5}$ (or something), and quote the value of the t-statistic.

This has been corrected.

Page 22, line 320: AMES->Ames

This has been corrected.

Page 23, line 329-334: Maybe add an example of these trajectories to one of the figure panels? (Sorry if I missed it)

An example of trajectory is present in figure 1, in gray.

Page 26 (lines 399-401): it is unclear whether signals pass through two separate nonlinearities (the one on line 400 and then the one in eq. 4 (line 401), or whether eq. 4 is just a restatement of the text.

We apologize for the confusion. There is only one non-linearity here, this is just a restatement.

We have clarified this in the text.

Page 28, line 437: Apparently some predicted PSTH's were negatively correlated with the model fit – the extent of these negative correlations should be mentioned here and in the figure legend for Figure 2F, as the authors indicate that it is suppressed in the graphical display. Presumably this only occurred for very peripheral bars, and the few points that are currently plotted on the axis.

We have replaced the figure to show these negative correlation values. As the reviewer can see, the values are still close to zero.

Page 32: Please clarify that information was calculated based on actual responses, not a model, as there would be a major caveat if the model response was used.

The information was calculated on the actual responses. We apologize if this was not clear.

We have clarified this in the text.

Reviewers' comments:

Reviewer #1 (Remarks to the Author):

The authors addressed most of my previous concerns with new data and analysis. However, some of these new results do not seem compatible with the proposed model. I think this is the matter of interpretation, rather than the data per se, but further clarification and reconciliation will be needed to present a model that is at least consistent across all data reported in the paper. Also, the revised manuscript has some errors in texts that need to be corrected.

=== Major points ===

(1) Figure 4:

line 158-159 and Figure 4C: It does not make sense to focus only on the positive weights here. The subunit model recovers the net effect of the positive (excitatory) and negative (inhibitory) inputs as the parameters (weights for each pathway), but cannot separate them. We should thus examine the weights as a whole. Indeed, Figure 4E-H clearly show substantial negative weights as well as positive ones.

line 159-163 and Figure 4D: Relative response size to ON or OFF stimulus does not necessarily inform the relative contribution of ON or OFF subunits. For example, there is a strong OFF response but virtually no ON response at the RF center in this OFF-type RGC (supp. fig. 1), but this does not mean that there is no contribution of ON subunits because they can drive the OFF response by "pull" mechanisms (and indeed this seems to be the case here as indicated by the negative weights at the RF center). Thus figure 4D alone does not necessarily support the dominance of the ON subunit. Again, I think what is needed is to properly examine the weights in the distant surround between ON and OFF subunits.

[A side note: I'm not sure if it is good for a change detector to have a dominant input from one pathway over the other. If ON pathway dominates with positive weights, for example, then the cells are sensitive to the appearance of ON stimulus or removal of OFF stimulus, but not to the appearance of OFF stimulus or removal of ON stimulus. Instead, it makes more sense (at least to me) if a change detector responds to any change, no matter what the stimulus polarity is.]

Lines 163-165: The claim in lines 163-165 does not seem to be well supported by the data in fig 4E, F. The subunit weights inside the RF center seemed to change in the example shown in Fig 4E-H. Please make a proper comparison of the weights (including both positive and negative weights) between single and combined bar conditions.

Line 165-167 and Figure 4I: Again, I think it is important to analyze the weights as a whole, instead of focusing only on the positive weights. Note there is a conflict between the plot (showing no negative values around zero distance) and its legend (saying "ON-subunit weights summed...").

Line 167-172: I think it is worth mentioning that there is no "push-pull" mechanisms for the distant circuits, while there is one in the RF center. This will highlight the difference between the center and distant circuits.

(2) Texture stimulus experiments:

Lines 108-115: Why did the subunit model perform so poorly here at a shorter distance (Supp. Fig.

3)? Is this related to the different nonlinearity used (power 4 in eq.9 for texture stimuli, as opposed to power 2 in eq.5 for bar stimuli)?

Lines 207-209: This is in contrast to the results in Supp. Fig 5F, showing that the suppression is independent of (or seemingly even negatively correlated with) the firing rate in response to the center stimulus. How do we reconcile this with the proposed model (and the results in Fig 5A and B)?

line 634: Please report the parameter values for eq.10 (just like the authors did for eq.6).

(3) Figure 6:

line 252. Figure 6E shows that central cells encode information about both position and speed, while distant cells encode information only about speed. Indeed, in absolute terms, the central cells encode as much information about speed as (or even more information than) the distant cells. Should we then interpret that these cells perform two computations with different spatial ranges, rather than switching the computations?

Figure 6C: how about the data for a central cell? The central cells seem to encode both speed and position (whereas the distant cells encode only speed).

(4) Pharmacology experiments

Lines 295-298: I disagree with the statement. Supp. fig. 7 shows that strychnine blocked "negative" weights for ON subunit in the receptive field center (panel B versus D). What we need to analyze here is to see if the weights for distant subunits (as in Figure 4) are suppressed, but not those at the RF center.

Also, it is worth discussing why the center response is not affected by strychnine. One possibility is that modeling captures some artifacts and thus the weights have nothing to do with biology. Another possibility is that suppression of the negative weights for the ON "pull" pathway and that for the ON "push" pathway somehow counteract each other to make the net effect negligible. Resolving this will require further careful examination with intracellular recordings, but this will be beyond the scope of this study.

Figure 7G: This is not a good example to support the suppression of surround response by picrotoxin. It instead shows that ON response to distant stimulation remains, and even OFF response arises after blocking GABA. Do you have a better example (that follows the population results as in the right panel of Fig 7H)?

Supplemental Figure 7: I am confused with the data shown here. Are they subunit pooling weights? If so, why the spatio-temporal profile is so different from those in Figure 4 (scale, peak latency, etc.)?

=== Minor points and suggestions ===

lines 27 and 67: Please change the word "linear" to "pseudo-linear" as in elsewhere.

line 29: Say "more nonlinearly" here for highlighting the difference from a "pseudo-linear" computation (as in the rebuttal).

line 48: State here that the bar width is 100 micron (or at the end of line 55) so that the readers can get a better idea on the spatial relationship between the stimulus and the receptive field size.

line 59: Move the phrase "Pearson correlation r" here from the line 62 to mention this for the first time use.

line 65: For clarification, better to say that "the bar moved within a region around 0.4mm wide".

line 106: State "(supp. fig. 7B and A, respectively)" to better specify the figure panels.

Line 185: The text needs to be reorganized. It says "suppression index estimated above", but the suppression index is mentioned here for the first time in the main text.

Line 194: It says "suppression index defined above", but it is not defined in the main text yet. I think it is better to refer to Eqs.12-13 in methods.

Line 299: please replace the word "showed" with "suggested."

Line 318-321: The authors can say more about the circuit (as in the rebuttal) and put more components in the schematics (Fig 7I) to summarize all the findings reported here. One possibility is that the disinhibition circuit is formed around the OFF "push" pathway at the RF center (Fig 7I). In this case, wide-field GABAergic AC inhibits Glycinergic AC that is involved in the RF (near) surround and inhibits OFF BC terminal presynaptically or GC postsynaptically (supp. fig. 7 AC). Another possibility is that the disinhibition circuit is formed around the ON ("pull") pathway at the RF center. In this case, wide-field GABAergic AC inhibits glycinergic AC that is mediating the "pull" inhibition from ON BC onto GC at the RF center (supp. fig. 7BD). Also, the authors could say that the inputs to those GABAergic ACs are likely from ON BCs (if indeed this is supported by the proper data analysis).

line 402 (typo); Should be figure 3, but not figure 4.

line 485 (typo): Should be "17-second" sequence, but not "17-seconds" sequence.

line 578: Remove the sentence saying "in Figure 2F, we set to zeros all negative Pearson coefficients." There are some negative values in Figure 2F.

lines 593, 596, 702: Correct the figure references to be the appropriate one.

lines 714, 720, 724: "micro" should be a Greek letter " μ ", and "Mol" should be replaced by "M".

Figure 2 and elsewhere: Correct the legend. The data PSTH is shown in black not in gray.

Figure 4B and elsewhere: Indicate what "****" indicates in the plot.

Figure 3G, inset: Indicate the time zero (when the spikes occur). I'm not fully convinced if we can classify this cell type as ON cells: the data suggest that, on average, the cells fires when the stimulus gets darker (from white to gray). Showing responses to a flash stimulus (as in supp. fig. 1) might be helpful. Also, it would be better to find the correspondence to the previous studies (as the author did for the OFF cell type).

Figure 7 legend (panel B and F) and elsewhere: Replace "same ~ than" with "same ~ as".

Supplemental Figure 1: The time axis labels are missing.

Supplemental Figure 2B: State that this is "the average pooling weights of the LN model", rather than the "average linear filter for the LN model." The LN model in this paper has in fact "LLN" architecture (but not the canonical LN where the L is often recovered by STA), and if I understand correctly, the figure panel here shows the second L process (while the first one is fixed). I would strongly suggest to modify the schematic in Figure 2A to avoid this confusion. It should show LLN cascade, and indicate which ones are fixed or optimized. Similarly, it would be useful to highlight the free parameters for the subunit model in the schematics (Figure 2D).

Supplemental Figure 3: State "the distance of the cell's receptive field center to texture border", but not "the distance of the cell to texture border." Or, for the consistency of the figure format, it might be better to use the distance to the center of texture stimulus and show the stimulus distribution overlaid (blue shade as in Figure 2F).

Supplemental Figure 4: correct the legend: "red" (but not "black") ellipse. Also show N in the legend (panel B).

Reviewer #2 (Remarks to the Author):

The authors have satisfactorily addressed my previous comments. Congratulations on a very nice study.

Reviewer #3 (Remarks to the Author):

The authors have undertaken an extensive revision of the manuscript, including new experiments and analyses, and they have thoughtfully considered my concerns. Many of my concerns have been fully addressed, and the study was (and even more so now is) technically excellent.

There are some remaining reservations: concerning matters of novelty (points a and b), and functional significance (d). With regard to novelty, I recognize that the matter is highly subjective and probably best left to the editor. With regard to (d), I would ask the authors to explicitly acknowledge the limitation detailed below.

Overall, I think that the new experiments and analyses make this a much stronger paper, and well worth publishing. The additional evidence supplied in response to points c and e is very convincing, and adds greatly to our conceptual understanding of mechanism. I think that these mechanistic insights should be emphasized, rather than the novelty of multiplexed computations per se or the computational model. Perhaps a change in title to "Mechanisms underlying multiplexed computations in ..."?

Details below:

a: the novelty of the point that cells can code for more than one feature

The authors acknowledge that previous work had already made this point, but state that the single-feature idea has recently been revived. They state that the novelty of their contribution to this issue is

that they are showing that information about multiple features can be transmitted simultaneously, or that different parts of the receptive field behave differently. But we already knew about center and surround, and that they have different dynamics. Bottom line, to me, is that this particular aspect of the work's conceptual advance is modest.

b: the novelty and advance in the proposed model

Here, the authors state that their model has an essential difference between that proposed for retinal ganglion cells by Shapley and colleagues in the 1970's – that in the earlier model, nonlinear subunits are also present in the center of the receptive field, but that in at least one cell type in their data, nonlinear subunits are not required in the center. Formally, this is indeed a difference – but it is worth noting that in the Shapley model, the linear receptive field elements dominated the subunit response in the center. There are also differences between how center and surround are weighted, in the authors' model compared to that of Shapley et al. So it is clear that the authors' models are different than the previous ones – though these differences seem more quantitative than conceptual.

One aspect of their response is not clear – they are contrasting a "LN" model (which is nonlinear), with cells that "remain nonlinear" – a confusing contrast since, after all, an LN model is nonlinear too.

c: whether the cells' behavior truly qualifies as a switching of modes

Their new evidence clearly shows that this is the case. I'm convinced.

d: the importance of the mode-switching behavior to decoding

If I understand correctly, they've addressed only half of the issue: they showed that the "distant" mode cannot decode position as well as the "central" mode, but they haven't asked whether the "central" mode can decode velocity as well as the "distant" mode (when looking at a population). The former is not surprising, since the central mode has greater spatial resolution than the distant mode. But the latter – whether a population "central" signals can substitute for the "distant" mode -- is the more critical and interesting point: from the point of view of population coding, is it necessary to have a special mode of processing in ganglion cells to signal speed, when they are already signaling position with good resolution? As mentioned above, I think that the authors should be explicit that they haven't addressed this point.

e: the pharmacologic dissection

This is a major and convincing addition to the paper.

The less-major points have all been fully addressed by further analyses, etc., including the critical issue of whether a simple threshold could account for their data.

One minor point that has arisen in the revision: the authors now introduce the term "pseudo-linear", and ascribe the term to my previous review. I can't find that term; I can only find the term "quasilinear", which I believe is more appropriate here. The LN model is approximately linear, not falsely linear.

Reviewers' comments:

Reviewer #1 (Remarks to the Author):

The authors addressed most of my previous concerns with new data and analysis. However, some of these new results do not seem compatible with the proposed model. I think this is the matter of interpretation, rather than the data per se, but further clarification and reconciliation will be needed to present a model that is at least consistent across all data reported in the paper. Also, the revised manuscript has some errors in texts that need to be corrected.

=== Major points ===

(1) Figure 4:

line 158-159 and Figure 4C: It does not make sense to focus only on the positive weights here. The subunit model recovers the net effect of the positive (excitatory) and negative (inhibitory) inputs as the parameters (weights for each pathway), but cannot separate them. We should thus examine the weights as a whole. Indeed, Figure 4E-H clearly show substantial negative weights as well as positive ones.

We agree with the reviewer that the influence of both positive and negative weights should be studied. We first emphasize that the subunit weights have a temporal structure (as can be seen in figure 4D,F): for both ON and OFF subunit, at the shortest time lag after the stimulus presentation, the subunit weights are positive. They become negative at longer time lags. Due to this temporal structure, it makes little sense to sum positive and negative weights together over all the time lags to compare them in different conditions. Most transient stimuli in the surround will trigger a first response due to the positive weights at short time lags, and then another response due to the negative weights, with a longer latency.

We therefore studied positive and negative subunit weights separately. We found that positive subunit weights were strongly decreased for both ON and OFF subunits, when a bar was presented in the center. On the contrary, negative weights were only weakly affected by the presence of the central bar, as shown by the two figures below, although we saw a small increase in ON negative weights.

The reduction of the response to distant stimuli is therefore mostly due to the decrease of the positive subunit weights.

We have modified figure 4 and the text (l. 157) to give a full account of these ON and OFF results for positive weights. We also show the negative weights in supplemental figure 5 and we mention them in the text (l. 165).

line 159-163 and Figure 4D: Relative response size to ON or OFF stimulus does not necessarily inform the relative contribution of ON or OFF subunits. For example, there is a strong OFF response but virtually no ON response at the RF center in this OFF-type RGC (supp. fig. 1), but this does not mean that there is no contribution of ON subunits because they can drive the OFF response by "pull" mechanisms (and indeed this seems to be the case here as indicated by the negative weights at the RF center). Thus figure 4D alone does not necessarily support the dominance of the ON subunit. Again, I think what is needed is to properly examine the weights in the distant surround between ON and OFF subunits.

We agree with the reviewer that the connection from the subunit weights to the net effect of a response to a flash is not direct. We have therefore removed this panel and statement from the text and provided a full analysis of ON and OFF subunits (see our reply to the previous point).

[A side note: I'm not sure if it is good for a change detector to have a dominant input from one pathway over the other. If ON pathway dominates with positive weights, for example, then the cells are sensitive to the appearance of ON stimulus or removal of OFF stimulus, but not to the appearance of OFF stimulus or removal of ON stimulus. Instead, it makes more sense (at least to me) if a change detector responds to any change, no matter what the stimulus polarity is.]

We agree with the reviewer that a perfect change detector should be equally sensitive to appearance and disappearance of ON and OFF stimuli. However, this is not what we observed, since our cells show much stronger response to an ON flash than an OFF one, and this is also what our model predicts.

We have therefore mentioned in the text that this computation is not a perfect change detector (l. 284).

Lines 163-165: The claim in lines 163-165 does not seem to be well supported by the data in fig 4E, F. The subunit weights inside the RF center seemed to change in the example shown in Fig 4E-H. Please make a proper comparison of the weights (including both positive and negative weights) between single and combined bar conditions.

We agree with the reviewer that the subunit weights inside the RF center may seem to change in previous figures 4E-H (now figures 4C-F). However, in the single distant bar condition, the bar spends very little time inside the receptive field of the cell. As a result, it is difficult to estimate the subunit weights inside the receptive field in this condition, and it is challenging to determine if the observed difference is due to a true change in the weights, or a lack of precision of the estimation of these weights. We found that making a proper comparison of these central weights between the two conditions was not possible. However, in this part of the text, our purpose was to quantify the suppression of the distant responses, so this claim was not necessary.

We have therefore taken back the claim that the central weights do not change and have mentioned why it is challenging to estimate them in the text (l. 637). In order to avoid possible confusions, we have also truncated the figures showing the subunit weights summed over all time lags (fig 4G-H and suppl. figure 5) so that they represent only the distant weights (further than 200 microns from RF center), because these are the only weights that can be estimated correctly in the two conditions (distant bar and two bars condition).

Line 165-167 and Figure 4I: Again, I think it is important to analyze the weights as a whole, instead of focusing only on the positive weights. Note there is a conflict between the plot (showing no negative values around zero distance) and its legend (saying "ON-subunit weights summed...").

As detailed in the reply to the first point raised by the reviewer, we have now examined the impact of both positive and negative weights.

We also corrected the legend and figure label of previous fig. 4I (now fig. 4G-H) to state explicitly that we are summing only positive weights.

Line 167-172: I think it is worth mentioning that there is no "push-pull" mechanisms for the distant circuits, while there is one in the RF center. This will highlight the difference between the center and distant circuits.

We thank the reviewer for the suggestion and have mentioned this in the text (l. 172). This is indeed an important difference between distant and central circuits.

(2) Texture stimulus experiments:

Lines 108-115: Why did the subunit model perform so poorly here at a shorter distance

(Supp. Fig. 3)? Is this related to the different nonlinearity used (power 4 in eq.9 for texture stimuli, as opposed to power 2 in eq.5 for bar stimuli)?

The reviewer is right that this is related to the different non-linearity. We could not predict the response to the randomly moving texture with a power 2 non-linearity, nor with a rectified linear one. In this case, the model cannot be used to approximate a LN model.

We have clarified this in the text (legend of supp. fig. 3).

Lines 207-209: This is in contrast to the results in Supp. Fig 5F, showing that the suppression is independent of (or seemingly even negatively correlated with) the firing rate in response to the center stimulus. How do we reconcile this with the proposed model (and the results in Fig 5A and B)?

We think this is not in contradiction because the two analyses are very different. On the one hand, in previous supplementary figure 5F (now fig. 6F), we have estimated the firing rate for each 16 ms time bin in response to the central bar alone. We then selected the time bin where the firing rate was above a given threshold and computed the suppression index on this subset of bins. This shows that the responses to the distant texture were equally suppressed in 16 ms time bins where the central texture triggered a moderate spiking, and in time bins where there was no spiking event in response to the central texture.

On the other hand, our results about the contrast dependency of the suppression indicate that the suppression is stronger when the contrast of the central bar increases. Here the suppression will be stronger when the response to the central bar averaged over a second is stronger.

These results are therefore not contradictory. They show that the suppression does not depend on the activity during the current 16 ms time bin, but suggest it may depend on the average activity over the last second. Note that this is compatible with our suggested mechanism of a gain control. Most gain control mechanisms described in the retina have a time constant of the order of hundreds of milliseconds, or seconds.

We have clarified this in the text (l. 222).

line 634: Please report the parameter values for eq.10 (just like the authors did for eq.6).

We have now reported these values (l. 674).

(3) Figure 6:

line 252. Figure 6E shows that central cells encode information about both position and speed, while distant cells encode information only about speed. Indeed, in absolute terms, the central cells encode as much information about speed as (or even more information than) the distant cells. Should we then interpret that these cells perform two computations with different spatial ranges, rather than switching the computations?

We agree that the two computations have clearly distinct spatial ranges, and that the central cells also encode speed. However, we think these two computations differ in

nature, since one of them can be approximated by a LN model, while the other cannot and can only be explained with the two layer model. This difference is related to the suggestion of the reviewer above that one of the computation (the central one) is probably generated by a push-pull circuit, while the other (the distant one) is not. The main reason why we emphasize the speed tuning of the distant cells is to demonstrate that they are selective to some aspect of the stimulus despite their lack of selectivity for the absolute position of the bar, but it is clear that they are not the only ones selective to speed.

For this reason we think this is more properly described as a switch between two computations of different natures, and we have clarified these arguments in the text (l. 280). In particular, we have clarified the fact that the distant cells are not the only one to code for speed (l. 259 and l. 280).

Figure 6C: how about the data for a central cell? The central cells seem to encode both speed and position (whereas the distant cells encode only speed).

We have added the same data for a central cell to clarify that central cells can also encode speed (new fig. 6C).

(4) Pharmacology experiments

Lines 295-298: I disagree with the statement. Supp. fig. 7 shows that strychnine blocked "negative" weights for ON subunit in the receptive field center (panel B versus D). What we need to analyze here is to see if the weights for distant subunits (as in Figure 4) are suppressed, but not those at the RF center.

First, we thank the reviewer for noticing a term mistake. We said in the text l. 295-298 that the blocker selectively suppressed the **positive** weights assigned to ON subunits, but what we meant is that it suppressed the **negative** weights assigned to ON subunits. Second, we agree that the structure of the main text could lead to confusion here. We would like to make clear the following points:

1) All the responses to the distant bar disappeared when adding strychnine. As a consequence, there was no correlation between the stimulus and the responses and all the subunit weights would be trivially zero for this condition.

2) As an independent observation, and as a secondary point not linked to the suppression of distant responses, we noticed that strychnine also had an effect on the central response to a randomly moving bar (but not to a flashed bar, see next point), by suppressing all the negative weights for ON and OFF subunits in the center.

We have made these two points more clearly in the text (l. 302 and l. 307), and we put them in separate paragraphs to avoid confusions.

Also, it is worth discussing why the center response is not affected by strychnine. One possibility is that modeling captures some artifacts and thus the weights have nothing to

do with biology. Another possibility is that suppression of the negative weights for the ON "pull" pathway and that for the ON "push" pathway somehow counteract each other to make the net effect negligible. Resolving this will require further careful examination with intracellular recordings, but this will be beyond the scope of this study.

Our results indicate that the response to the **flashed** bar in the center was maintained in the presence of strychnine. However, when fitting the subunit model to the **randomly moving bar** condition, the subunit weights in the **receptive field center** were affected by strychnine: negative weights of the ON and OFF subunits disappeared. These results show that the response to the flashed central bar is maintained, but with a more complex stimulus like the randomly moving bar, we could see some significant changes in the responses inside the center. There is no incompatibility in these two observations because the flashed bar elicits only OFF responses and thus should not be affected by the disappearance of ON negative weights in the center. Moreover, the OFF negative weights are not present in the center but only in the surround so their disappearance should not affect the response either.

We have clarified this in the text (l. 307).

Figure 7G: This is not a good example to support the suppression of surround response by picrotoxin. It instead shows that ON response to distant stimulation remains, and even OFF response arises after blocking GABA. Do you have a better example (that follows the population results as in the right panel of Fig 7H)?

We have displayed another, better example. However, it is worth noting that, as explained previously (l. 324), we had to use a limited concentration of picrotoxin to avoid triggering large oscillations in the retinal activity. As a result, we could not expect a complete suppression of the distant responses. In many cases, the response to distant bar was decreased but a small response remained.

We have mentioned this in the text (l. 766).

Supplemental Figure 7: I am confused with the data shown here. Are they subunit pooling weights? If so, why the spatio-temporal profile is so different from those in Figure 4 (scale, peak latency, etc.)?

We thank the reviewer for noticing this discrepancy. There was a mistake in the estimation of the subunit weights that we have corrected. The estimation of the subunit weights are a bit noisier here than for the other experiments because the length of the recordings were shortened for the pharmacological experiment.

=== Minor points and suggestions ===

lines 27 and 67: Please change the word "linear" to "pseudo-linear" as in elsewhere.

Following the suggestion of reviewer 3 we have termed the LN computation "quasilinear" everywhere in the paper.

line 29: Say "more nonlinearly" here for highlighting the difference from a "pseudo-linear" computation (as in the rebuttal).

This has been changed.

line 48: State here that the bar width is 100 micron (or at the end of line 55) so that the readers can get a better idea on the spatial relationship between the stimulus and the receptive field size.

This has been changed.

line 59: Move the phrase "Pearson correlation r " here from the line 62 to mention this for the first time use.

This has been changed.

line 65: For clarification, better to say that "the bar moved within a region around 0.4mm wide".

This has been changed.

line 106: State "(supp. fig. 7B and A, respectively)" to better specify the figure panels.

This has been changed.

Line 185: The text needs to be reorganized. It says "suppression index estimated above", but the suppression index is mentioned here for the first time in the main text.

We have corrected this.

Line 194: It says "suppression index defined above", but it is not defined in the main text yet. I think it is better to refer to Eqs.12-13 in methods.

We have corrected this.

Line 299: please replace the word "showed" with "suggested."

We have corrected this.

Line 318-321: The authors can say more about the circuit (as in the rebuttal) and put more components in the schematics (Fig 7I) to summarize all the findings reported here. One possibility is that the disinhibition circuit is formed around the OFF "push" pathway at the RF center (Fig 7I). In this case, wide-field GABAergic AC inhibits Glycinergic AC that is involved in the RF (near) surround and inhibits OFF BC terminal presynaptically or GC postsynaptically (supp. fig. 7 AC). Another possibility is that the disinhibition circuit is formed around the ON ("pull") pathway at the RF center. In this case, wide-field GABAergic AC inhibits glycinergic AC that is mediating the "pull" inhibition from ON BC onto GC at the RF center (supp. fig. 7BD). Also, the authors could say that the inputs to

those GABAergic ACs are likely from ON BCs (if indeed this is supported by the proper data analysis).

We agree with the reviewer that we could have said more about this in the paper. We have added details about these possible circuits in the text (l. 339). We decided not to include them in the circuit of the schematics to avoid crowding this panel.

line 402 (typo); Should be figure 3, but not figure 4.

We have corrected this.

line 485 (typo): Should be "17-second" sequence, but not "17-seconds" sequence.

We have corrected this.

line 578: Remove the sentence saying "in Figure 2F, we set to zeros all negative Pearson coefficients." There are some negative values in Figure 2F.

We have corrected this.

lines 593, 596, 702: Correct the figure references to be the appropriate one.

We have corrected this.

lines 714, 720, 724: "micro" should be a Greek letter " μ ", and "Mol" should be replaced by "M".

We have corrected this.

Figure 2 and elsewhere: Correct the legend. The data PSTH is shown in black not in gray.

We have corrected this.

Figure 4B and elsewhere: Indicate what "****" indicates in the plot.

We have corrected this.

Figure 3G, inset: Indicate the time zero (when the spikes occur). I'm not fully convinced if we can classify this cell type as ON cells: the data suggest that, on average, the cells fires when the stimulus gets darker (from white to gray). Showing responses to a flash stimulus (as in supp. fig. 1) might be helpful. Also, it would be better to find the correspondence to the previous studies (as the author did for the OFF cell type).

We apologized for the mistake: we put the temporal profile of the first OFF type instead of this type. We have corrected this. The temporal profiles are biphasic but have a clear ON peak.

Below is also an example of response of one of these cells to a flash, showing a clear response when the full field stimulus switches from black to white (at 0.5 s), while the switch from white to black (at 0 s) triggers a suppression of spontaneous activity. We

have mentioned this result in the text. We have also found the most likely correspondence with previous studies and put it in the text (l. 124).

Figure 7 legend (panel B and F) and elsewhere: Replace "same ~ than" with "same ~ as".

We have corrected this.

Supplemental Figure 1: The time axis labels are missing.

We have corrected this.

Supplemental Figure 2B: State that this is "the average pooling weights of the LN model", rather than the "average linear filter for the LN model." The LN model in this paper has in fact "LLN" architecture (but not the canonical LN where the L is often recovered by STA), and if I understand correctly, the figure panel here shows the second L process (while the first one is fixed). I would strongly suggest to modify the schematic in Figure 2A to avoid this confusion. It should show LLN cascade, and indicate which ones are fixed or optimized. Similarly, it would be useful to highlight the free parameters for the subunit model in the schematics (Figure 2D).

We apologize for the confusion. What is shown here is the filter of a LN model. It is true that we estimated the LN model using a basis to reduce the number of parameters and be able to learn the model with a limited amount of data, and this is common practice in the literature (e.g. Pillow et al, 2008). We then projected the results back in the filter basis to obtain the "complete" filter of the LN model, which is much easier to interpret. Following the reviewer suggestion we modified figure 2D to highlight the learned parameters in the subunit model.

Supplemental Figure 3: State "the distance of the cell's receptive field center to texture border", but not "the distance of the cell to texture border." Or, for the consistency of the figure format, it might be better to use the distance to the center of texture stimulus and show the stimulus distribution overlaid (blue shade as in Figure 2F).

We corrected this by stating "the distance of the cell's receptive field center to texture border", but we prefer to keep the distance to the texture border as a reference rather than to the texture center, since the texture covers a broad area and its center position is not very informative.

Supplemental Figure 4: correct the legend: "red" (but not "black") ellipse. Also show N in the legend (panel B).

This has been corrected.

Reviewer #2 (Remarks to the Author):

The authors have satisfactorily addressed my previous comments. Congratulations on a very nice study.

We thank the reviewer for his positive comment.

Reviewer #3 (Remarks to the Author):

The authors have undertaken an extensive revision of the manuscript, including new experiments and analyses, and they have thoughtfully considered my concerns. Many of my concerns have been fully addressed, and the study was (and even more so now is) technically excellent.

There are some remaining reservations: concerning matters of novelty (points a and b), and functional significance (d). With regard to novelty, I recognize that the matter is highly subjective and probably best left to the editor. With regard to (d), I would ask the authors to explicitly acknowledge the limitation detailed below.

Overall, I think that the new experiments and analyses make this a much stronger paper, and well worth publishing. The additional evidence supplied in response to points c and e is very convincing, and adds greatly to our conceptual understanding of mechanism. I think that these mechanistic insights should be emphasized, rather than the novelty of multiplexed computations per se or the computational model. Perhaps a change in title to "Mechanisms underlying multiplexed computations in ..."?

We understand the suggestion of the reviewer. However, we think that, if we were to choose this title, a significant part of the readership would expect a detailed study of the retinal circuits underlying the distant responses and their suppression, using intracellular recordings of different cell types, which is not the topic of this paper. We kept the title and discussed the novelty aspect below and in the paper.

Details below:

a: the novelty of the point that cells can code for more than one feature

The authors acknowledge that previous work had already made this point, but state that the single-feature idea has recently been revived. They state that the novelty of their contribution to this issue is that they are showing that information about multiple features can be transmitted simultaneously, or that different parts of the receptive field behave differently. But we already knew about center and surround, and that they have different dynamics. Bottom line, to me, is that this particular aspect of the work's conceptual advance is modest.

We understand why this part of the paper may seem like a modest conceptual advance. However, we think it is important for our study to re-demonstrate this point, otherwise the rest of the paper, especially the mode switching, cannot be understood. Furthermore, as mentioned previously, we think that it is important to demonstrate this point in a case where a single cell type is isolated and many cells of the same type are recorded simultaneously.

We have clarified this in the text (l. 376).

b: the novelty and advance in the proposed model

Here, the authors state that their model has an essential difference between that proposed for retinal ganglion cells by Shapley and colleagues in the 1970's – that in the earlier model, nonlinear subunits are also present in the center of the receptive field, but that in at least one cell type in their data, nonlinear subunits are not required in the center. Formally, this is indeed a difference – but it is worth noting that in the Shapley model, the linear receptive field elements dominated the subunit response in the center. There are also differences between how center and surround are weighted, in the authors' model compared to that of Shapley et al. So it is clear that the authors' models are different than the previous ones – though these differences seem more quantitative than conceptual.

We agree that conceptually, the subunit model we used may seem quite similar to the one of Shapley et al, although there are some clear quantitative differences. However, we also show that the type of ganglion cells determine whether the non linear subunits or the linear component prevail in the center. This systematic difference due to the cell type was not described in Shapley et al. We think the re-assessment of the non-linearities in the case where cell types are well defined and isolated is novel compared to Shapley et al, where the final model (fig 10) showed the presence of non-linear subunits and linear components both in the center and the surround. Finally, our results also show how the linear computation in the center emerges from the organization of the ON and OFF subunits in a push-manner, where negative weights for ON correspond to positive weights for the OFF subunit, so that the total sum has a linear relation to the stimulus. We think this is an interesting aspect that was not reported in Shapley et al.

We have clarified this in the text (l. 376).

One aspect of their response is not clear – they are contrasting a “LN” model (which is nonlinear), with cells that “remain nonlinear” – a confusing contrast since, after all, an LN model is nonlinear too.

We apologize for the confusion. Here we wanted to contrast the quasilinear LN model and the two layer model we used to predict the distant responses.

We have clarified this in the text (l. 169).

c: whether the cells' behavior truly qualifies as a switching of modes

Their new evidence clearly shows that this is the case. I'm convinced.

d: the importance of the mode-switching behavior to decoding

If I understand correctly, they've addressed only half of the issue: they showed that the "distant" mode cannot decode position as well as the "central" mode, but they haven't asked whether the "central" mode can decode velocity as well as the "distant" mode (when looking at a population). The former is not surprising, since the central mode has greater spatial resolution than the distant mode. But the latter – whether a population "central" signals can substitute for the "distant" mode ----- is the more critical and interesting point: from the point of view of population coding, is it necessary to have a special mode of processing in ganglion cells to signal speed, when they are already signaling position with good resolution? As mentioned above, I think that the authors should be explicit that they haven't addressed this point.

We apologize if this was not clear before: it is possible to decode speed from central ganglion cells. Central cells give access to the position of the bar with a good resolution and therefore also give a good representation of speed. We acknowledge that we have not found a feature that could not be decoded by central cells but only by distant ones. We have focused on position and speed because it allows demonstrating that the lack of sensitivity to position of distant cells does not mean they are generally less sensitive to any feature of the stimulus. But it does not mean that these distant cells are the only ones coding for speed: our previous analysis showed that they don't carry more information about speed than central cells. We therefore acknowledged in the text that we have not found a feature that could not be decoded by the central cells (l. 280).

However, we think that cells are not defined only by the features they are informative about – they are also defined by their invariance to some changes. Here the functional significance of distant cells might reside in their relative invariance to position, which might be useful to build global change detectors in downstream areas.

e: the pharmacologic dissection

This is a major and convincing addition to the paper.

The less-major points have all been fully addressed by further analyses, etc., including the critical issue of whether a simple threshold could account for their data.

One minor point that has arisen in the revision: the authors now introduce the term "pseudo-linear", and ascribe the term to my previous review. I can't find that term; I can only find the term "quasilinear", which I believe is more appropriate here. The LN model is approximately linear, not falsely linear.

We thank the reviewer for this suggestion and have used this term in the paper.

Reviewers' comments:

Reviewer #1 (Remarks to the Author):

The authors have addressed most of my previous concerns, but not all of them. As detailed below, a critical point on the circuit mechanisms underlying the switch between the two computations remains and needs to be clarified.

=== major point ===

Figure 4: I disagree with the authors' argument.

Lines 157-158: "For both ON and OFF distant subunits, at the shortest time lag after the stimulus presentation, the subunit weights were positive."

At least an example in Figure 4E shows negative weights at the shortest time lag. (By the way, it is not clear exactly what time points "the shortest time lag" refers to.) Also, if the shortest time lag is important here, it does not make sense to sum the weights over all time lags.

Lines 163-167: "positive subunit weights were strongly decreased [while] negative weights were only weakly affected by the presence of the central bar"

This is not supported by the data: the difference is ~ 0.005 for the ON negative weights at 600-800 microns from the RF center, while ~ 0.002 for the ON positive weights. (Also, it is not clear to me exactly at what distance the "distant subunits" are.) The scale is so different between the positive and negative weights (and also between ON and OFF subunits; this comes back to my original comments on their contributions), and thus one cannot make an eye-ball comparison here. Please show everything on the same scale.

For clarification, I would suggest the following two analyses.

1) Simply take the difference of the weights under two conditions, and show the results for ON and OFF subunits on the same scale. Specifically, for the ON subunits, subtract the spatio-temporal weights in C from those in D, and show this difference in a new panel next to D. Similarly, show the difference of the OFF subunit weights next to F. Then, you can take an average of the difference over a given time window (say, time lag between 0 and 200 ms) to compare the contributions of the ON and OFF subunits (at different locations).

2) Pharmacologically block the ON pathway to resolve the contributions of the ON and OFF pathways. For example, repeat the experiments in 2-amino-4-phosphono-butyrates, a metabotropic glutamate receptor agonist that blocks transmission to ON bipolar cells. If indeed the response to the distant stimulation is suppressed as well as the ON subunit weights (while those of the OFF subunits remain unaffected), then this will give a strong experimental evidence favoring the circuit mechanisms proposed by the authors.

=== minor points and suggestion ===

line 7 in abstract: should be "quasi-linearly" but not "linearly".

line 27 (typo): should be "overlaps" but not "overlap".

lines 124-126: I think it reads better if the authors change the sentence as follows:

We analyzed another type of retinal ganglion cells in our data set whose receptive fields clearly tiled the visual field (fig. 3G): namely, ON cells that probably correspond to ON transient type 18a or 18b in (Baden et al, 2016) as they responded transiently to an ON full field flash but not to an OFF one.

Lines 339-348: I still think it is useful to show a circuit diagram that summarizes the findings and the proposed mechanisms (maybe in Figure 8 if Figure 7 is too crowded). It indeed helps readers appreciate the novelty aspect of the paper (as the reviewer 3 pointed out) because we all want to know not just "what" the retina does but also "how" the retina does so.

Figure 1D legend (and elsewhere): should be "SEM" but not "SE". Please do not interchange the abbreviations in the same manuscript.

Figure 3 legend: "Filled arrows correspond to learned weights of the model." It would be better to fill the ellipses (linear filters) / squares (nonlinearity), rather than the arrows, to indicate which part of the model is fitted. And please do the same for both Fig 3A and 3D.

Figure 3E legend: should be "to central stimulation (as in B)" but not "(as in B and C) to central stimulation".

Figure 3F legend: should be "to distant stimulation (as in C)" but not "(as in B and C) to distant stimulation".

Figure 3G: the scale bar should be " μm ", but not "microns"

Figure 4G: use the same blue color for the plot and the figure label (with central bar)

Supplemental Figure 1A: show "Gray area" in the figure panel as described in the legend.

Supplemental Figure 3: Please correct the legend. The data PSTH is shown in black not in gray.

Supplemental Figure 5: should be " μm " but not "microns".

Supplemental Figure 6: please indicate that "****" indicates $p < 0.001$.

Reviewer #3 (Remarks to the Author):

The authors have now made focal edits that contain appropriate caveats and clarifications. Any remaining concerns are definitely in the domain of subjective/opinion. I have no further concerns.

Reviewers' comments:

Reviewer #1 (Remarks to the Author):

The authors have addressed most of my previous concerns, but not all of them. As detailed below, a critical point on the circuit mechanisms underlying the switch between the two computations remains and needs to be clarified.

=== major point ===

Figure 4: I disagree with the authors' argument.

Lines 157-158: "For both ON and OFF distant subunits, at the shortest time lag after the stimulus presentation, the subunit weights were positive."

At least an example in Figure 4E shows negative weights at the shortest time lag. (By the way, it is not clear exactly what time points "the shortest time lag" refers to.) Also, if the shortest time lag is important here, it does not make sense to sum the weights over all time lags.

Lines 163-167: "positive subunit weights were strongly decreased [while] negative weights were only weakly affected by the presence of the central bar"

This is not supported by the data: the difference is ~ 0.005 for the ON negative weights at 600-800 microns from the RF center, while ~ 0.002 for the ON positive weights. (Also, it is not clear to me exactly at what distance the "distant subunits" are.) The scale is so different between the positive and negative weights (and also between ON and OFF subunits; this comes back to my original comments on their contributions), and thus one cannot make an eye-ball comparison here. Please show everything on the same scale.

We have put the filters of figure 4 at the same scale. To enhance the visibility of distant subunit weights, and because central weights are not relevant here, in this new panel the central weights are saturated. We have also performed the analyses suggested below.

For clarification, I would suggest the following two analyses.

1) Simply take the difference of the weights under two conditions, and show the results for ON and OFF subunits on the same scale. Specifically, for the ON subunits, subtract the spatio-temporal weights in C from those in D, and show this difference in a new panel next to D. Similarly, show the difference of the OFF subunit weights next to F. Then, you can take an average of the difference over a given time window (say, time lag between 0 and 200 ms) to compare the contributions of the ON and OFF subunits (at different locations).

We thank the reviewer for this suggestion. We have performed this analysis (now supplementary figure 5 and figure 4 G-H). Subtracting the filters shows a decrease of the ON subunit weights everywhere. For the OFF subunit weights, a similar effect appears, but there is also an increase of the weights at the space-time location where these weights were negative. Between 0 and 200 ms, there is an overall decrease of the subunit weights, for both ON and OFF subunits.

We have described these results in the text. Since these analyses are redundant with, but clearer (and easier to understand) than the previous separation between positive and negative weights, we have removed this previous analysis from the manuscript.

2) Pharmacologically block the ON pathway to resolve the contributions of the ON and OFF pathways. For example, repeat the experiments in 2-amino-4-phosphonobutyrate, a metabotropic glutamate receptor agonist that blocks transmission to ON bipolar cells. If indeed the response to the distant stimulation is suppressed as well as the ON subunit weights (while those of the OFF subunits remain unaffected), then this will give a strong experimental evidence favoring the circuit mechanisms proposed by the authors.

We have performed this additional experiment. We found that distant responses were mostly (although not completely) suppressed when adding LAP4 to the bath (supplementary figure 9 and text). From this we conclude that the distant responses are dominated by a circuit that is stimulated by ON bipolar cells.

Similar to strychnine, we could not obtain subunit filters, neither for the ON nor for the OFF subunit weights, under LAP-4. One possible explanation could be that the residual distant responses due to a putative OFF pathway were buried in the noise, and therefore impossible to find when fitting the model on the limited amount of data acquired during an experiment with pharmacological manipulation.

Another explanation is that the correspondence between the components of this model and the elements of the circuit is not straightforward, especially for distant subunits. Our subunit model is a good functional model for the processing performed by these ganglion cells, since it predicts well how these cells respond to the stimuli. However, while our subunits are inspired by ON and OFF bipolar cells, we cannot claim that there is a perfect correspondence between ON subunit and ON bipolar cells, especially for distant subunits. The difficulty of mapping components of a phenomenological model and cells of a circuit has been the subject of many previous studies. Here the distant subunit weights reflect the impact of a circuit composed of two amacrine cells. A simple coefficient in the phenomenological model thus corresponds to a relay of two amacrine cells. This is an example showing the difficulty of trying to map a phenomenological model to a circuit, but many other examples exist in the literature.

We have added these results in the text and have emphasized that the correspondence between bipolar cells and subunits is not demonstrated.

=== minor points and suggestion ===

line 7 in abstract: should be "quasi-linearly" but not "linearly".

We have corrected this.

line 27 (typo): should be "overlaps" but not "overlap".

We have corrected this.

lines 124-126: I think it reads better if the authors change the sentence as follows: We analyzed another type of retinal ganglion cells in our data set whose receptive fields clearly tiled the visual field (fig. 3G): namely, ON cells that probably correspond to ON transient type 18a or 18b in (Baden et al, 2016) as they responded transiently to an ON full field flash but not to an OFF one.

We thank the reviewer for this better formulation and we have adopted it in the text.

Lines 339-348: I still think it is useful to show a circuit diagram that summarizes the findings and the proposed mechanisms (maybe in Figure 8 if Figure 7 is too crowded). It indeed helps readers appreciate the novelty aspect of the paper (as the reviewer 3 pointed out) because we all want to know not just "what" the retina does but also "how" the retina does so.

We have now included a more detailed circuit diagram that summarizes the finding.

Figure 1D legend (and elsewhere): should be "SEM" but not "SE". Please do not interchange the abbreviations in the same manuscript.

We have corrected this.

Figure 3 legend: "Filled arrows correspond to learned weights of the model." It would be better to fill the ellipses (linear filters) / squares (nonlinearity), rather than the arrows, to indicate which part of the model is fitted. And please do the same for both Fig 3A and 3D.

In the subunit model the linear filter which pools the subunit outputs was fitted, while the subunit parameters were held fixed. It is therefore correct to say that the arrows are the learned part of the model.

We have also filled the arrows in fig 3A as suggested.

Figure 3E legend: should be "to central stimulation (as in B)" but not "(as in B and C) to central stimulation".

We have corrected this.

Figure 3F legend: should be "to distant stimulation (as in C)" but not "(as in B and C) to distant stimulation".

No, our statement is correct because we used here the same cell in two conditions (central bar and distant bar stimulation).

Figure 3G: the scale bar should be " μm ", but not "microns"

This has been corrected.

Figure 4G: use the same blue color for the plot and the figure label (with central bar)

This panel has been changed (see above).

Supplemental Figure 1A: show "Gray area" in the figure panel as described in the legend.

This has been corrected.

Supplemental Figure 3: Please correct the legend. The data PSTH is shown in black not in gray.

We have corrected this.

Supplemental Figure 5: should be " μm " but not "microns".

This figure has been changed (see above).

Supplemental Figure 6: please indicate that "****" indicates $p < 0.001$.

We have corrected this.

Reviewer #3 (Remarks to the Author):

The authors have now made focal edits that contain appropriate caveats and clarifications. Any remaining concerns are definitely in the domain of subjective/opinion. I have no further concerns.

REVIEWERS' COMMENTS:

Reviewer #1 (Remarks to the Author):

The authors have addressed all my concerns. This is an excellent work that is definitely worth publishing.